# Enhancing Maritime Trajectory Forecasting via H3 Index and Causal Language Modelling (CLM)

**Nicolas Drapier**[*]  *nicolas.drapier@etu.univ-orleans.fr, nicolas.drapier@sas-impact.fr*
*PRISME Laboratory, SAS Impact*
*University of Orléans, Orléans, France*

**Aladine Chetouani**[†]  *aladine.chetouani@univ-orleans.fr*
*PRISME Laboratory*
*University of Orléans, Orléans, France*

**Aurélien Chateigner**[‡]  *aurelien.chateigner@sas-impact.fr*
*SAS Impact*
*Orléans, France*

**Reviewed on OpenReview:** *https://openreview.net/forum?id=tIfS6jyO9f*

## Abstract

The prediction of ship trajectories is a growing field of study in artificial intelligence. Traditional methods rely on the use of LSTM, GRU networks, and even Transformer architectures for the prediction of spatio-temporal series. This study proposes a viable alternative for predicting these trajectories using only GNSS positions. It considers this spatio-temporal problem as a natural language processing problem. The latitude/longitude coordinates of AIS messages are transformed into cell identifiers using the H3 index. Thanks to the pseudo-octal representation, it becomes easier for language models to learn the spatial hierarchy of the H3 index. The method is qualitatively compared to a classical Kalman filter and quantitatively to Seq2Seq and TrAISformer models. The Fréchet distance is introduced as the main evaluation metric for these comparisons. We show that it is possible to predict ship trajectories quite precisely up to 8 hours ahead with 30 minutes of context, using solely GNSS positions, without relying on any additional information such as speed, course, or external conditions — unlike many traditional methods. We demonstrate that this alternative works well enough to predict trajectories worldwide.

## 1 Introduction

Predicting ship trajectories is an essential task for maritime stakeholders, encompassing economic, security, and logistical considerations. Accurate trajectory prediction plays a pivotal role in optimising shipping routes, ensuring maritime safety, and managing resources efficiently. However, this endeavour has posed several challenges due to the vast amount of trajectory data generated in real-time and the intricate interplay of spatial and temporal factors.

Traditionally, Long Short-Term Memory (LSTM) (Hochreiter & Schmidhuber, 1997) and Gated Recurrent Units (GRU) (Cho et al., 2014) networks have been employed to model sequential and temporal data, and many researchers have tried to adapt these recurrent neural network (RNN) architectures to the spatio-temporal domain. While these RNN-based approaches have demonstrated success in various applications

---

[*]Principal author, Conceptualisation, Methodology, Software, Validation, Formal analysis, Investigation, Resources, Data Curation, Writing – original draft, Visualisation

[†]Writing – review & editing, Visualisation, Supervision, Project administration

[‡]Conceptualisation, Methodology, Validation, Formal analysis, Writing – review & editing, Visualisation, Supervision, Project administration

(Connor et al., 1994; Assaad et al., 2008; Salinas et al., 2020; Khan et al., 2020), they typically neglect the crucial spatial component inherent in ship trajectories, such as the geographical coordinates and the intricate relationships between waypoints in a trajectory.

More recently, the introduction of the Transformer architecture by Vaswani et al. (2017) has sparked a revolution in different fields, such as the Natural Language Processing (NLP) (Brown et al., 2020; Radford et al., 2019; Devlin et al., 2018) and time-series prediction (Nie et al., 2022; Zeng et al., 2022; Liu et al., 2023; Wu et al., 2022), democratising the development of state-of-the-art models due to their popularity, scalability, and ease of use. Models like Informer (Zhou et al., 2021) and Autoformer (Wu et al., 2021) have shown exceptional prowess in capturing long-term dependencies and intricate relationships within temporal data, making them valuable tools in the realm of time-series prediction.

However, despite their successes in temporal modelling, these models based on the Transformer architecture have not been adequately tailored to address the spatial nuances inherent in ship trajectories, limiting their applicability in the maritime domain. This research effort seeks to bridge this gap by exploring novel approaches that combine the strengths of Transformer-based architectures with spatial awareness, offering the potential to revolutionise ship trajectory prediction in real-time industrial applications. In doing so, we aim to address the limitations of existing models, making maritime operations more efficient, secure, and economically viable.

In this research, we adopt a Causal Language Model (CLM) due to its ability to effectively handle sequential dependencies, a crucial aspect of ship trajectory prediction. The auto-regressive nature of CLMs aligns with the temporal progression of vessel movements, where each position is inherently influenced by prior locations. By leveraging the capacity of CLMs to predict future tokens based on preceding ones, we ensure that the spatial and temporal continuity of trajectories is accurately captured.

We believe that combining Transformer-based models with better spatial representation can enhance the accuracy of ship trajectory prediction. In maritime trajectory prediction, accurately representing Global Navigation Satellite System (GNSS) coordinates is challenging due to the inherent precision of GNSS measurements. Even a minor error of one hundredth of a degree in latitude consistently translates to a difference of approximately 1.11 kilometres, while in longitude, the impact varies considerably based on Earth's location. For instance, at the equator, it approximates 1.11 kilometres, but at 80° longitude, it diminishes to around 200 metres. These variations lead to profound disparities in actual locations, making it difficult to increase the dimensionality of GNSS positions using an autoencoder-based approach.

In light of these considerations, we set out to explore an alternative solution: the discretisation method utilising Uber's Hexagonal Hierarchical Spatial Index (H3) (Uber, 2018). H3 is a geospatial indexing system that partitions the world into hexagonal cells. H3 cells are written to 64 bits and can be represented in hexadecimal format, providing a creative solution to the challenges associated with the continuous representation of GNSS coordinates. While the concept of partitioning trajectories into hexagonal cells has already been explored by Spadon et al. (2024), their approach does not strictly adhere to Uber's original H3 method. Instead, they propose a similar discretisation technique, which inspired us to refine the use of H3 for our purposes. Our central aim with this discretisation approach is to reconfigure a ship's trajectory from Automatic Identification System (AIS) [1] messages (Organization, 2015) into a structured narrative, akin to the composition of words in a sentence. By doing so, we effectively counteract the potential detrimental impacts of minor coordinate inaccuracies, and we expect to benefit from the ability of Transformer architectures to understand the relationships between tokens.

Given the limited usefulness of the initial 12 bits of the H3 cells for improving our understanding of space, we propose an innovative pseudo-octal format. This representation is a cornerstone of our study. By tailoring this and integrating it with a customised tokenisation scheme, we have constructed a tokeniser vocabulary of just 270 tokens. This streamlined encoding not only optimises the computational load but also facilitates the Transformer models' spatial comprehension. Together, this pseudo-octal representation and tokenisation allow the models to grasp the structure of ship trajectories, offering a significant advantage in spatial pattern recognition within our Transformer-based architecture.

---

[1] The *Automatic Identification System* is an automatic tracking system used to monitor and locate ships in real time. It uses Very High Frequency (VHF) transceivers to transmit data such as the vessel's identifier, position, speed, and direction.

To clarify the role of H3 in our method, it is important to note that while H3 plays a crucial role in structuring our data, it is not the core innovation of our approach. H3 serves as a tool to add hierarchical spatial structure to the continuous GNSS coordinates, enabling us to represent ship trajectories as sequences of tokens. This structured representation is essential for our CLM to process the data effectively. Our primary focus lies in leveraging Large Language Models (LLMs) to process time-series data by representing these sequences in a "natural" language with task-specific grammar, where the grammar is defined by the spatial structure provided by H3. This approach allows us to treat trajectory prediction as a language modelling task, where the model learns to predict the future sequence based on the patterns and relationships it identifies in the data.

To analyse the results, we propose utilising the Fréchet distance (Fréchet, 1957). The rationale behind this choice lies in its ability to capture the similarity between two trajectories while considering both spatial proximity and sequential order of points. The Fréchet distance, by comparing the geometric similarity between the actual and predicted trajectories, serves as a stringent measure of predictive fidelity.

In the forthcoming sections, we delve into the methodologies adopted, with a primary emphasis on the utilisation of the H3 index for spatial representation. We elaborate on the rationale behind selecting H3 at a zoom level of 10, which directly influences the granularity of the discretised ship trajectories. This choice plays a critical role in the quality of predictions, as the level of precision in the H3 index determines the amount of zigzagging introduced into a trajectory after discretisation. A finer resolution could capture more detailed movements, but may also increase noise, whereas a coarser resolution might miss important spatial nuances.

Furthermore, we present experimental results and validate our approach against existing methods, demonstrating the promise of integrating Transformer architectures with spatial awareness within the maritime domain.

## 2  Background

GNSS play a pivotal role in modern artificial intelligence, particularly in applications related to Geospatial Intelligence (GEOINT). GEOINT, as defined by the USA, refers to the analysis and exploitation of geospatial information to gain insights and support joint operations (Sharp, 2011; Clark, 2020). GNSS, which includes systems like GPS, GLONASS, and Galileo provides essential geospatial data that underpins these intelligence operations.

A complementary technology in the maritime domain is the Automatic Identification System (AIS) (Organization, 2015), which is increasingly intertwined with GNSS systems. AIS transponders on ships broadcast their GNSS-derived positions, allowing for real-time tracking and identification. This integration has revolutionised maritime operations, enabling vessel tracking, collision avoidance, and efficient port management.

However, AIS is limited due to inherent characteristics. The protocol's reliance on VHF radio transmissions limits its range, making it susceptible to signal loss in remote or congested areas and not always accurate due to delays in transmitting the data. Additionally, while AIS data provides valuable information, it may not always reflect the full scope of a vessel's activities, particularly in scenarios involving illicit or evasive behaviour.

One of the fundamental challenges in utilising GNSS and AIS data is the open nature of the ocean. Unlike terrestrial environments, where fixed road networks and landmarks facilitate navigation, the vast and unpredictable expanse of the sea makes it difficult to determine a vessel's trajectory with absolute certainty. The dynamic and often erratic nature of maritime operations poses significant obstacles to generating accurate predictions.

In the field of natural language processing (NLP) research, Transformer architectures with progressively extended model capabilities are at the heart of research. They are originally designed for NLP applications, and they have undergone substantial improvements in order to optimise their efficiency in the processing of sequential data. Recent progress in this line of research has been directed towards improving their ability to handle extended sequences efficiently (Xiong et al., 2021; Beltagy et al., 2020; Child et al., 2019; Press et al.,

2021; Dai et al., 2019; Su et al., 2024), making transformers an appropriate choice for the complex analysis of large datasets on maritime trajectories.

## 3 Related Work

In recent years, the field of vessel trajectory prediction and maritime safety has witnessed significant developments, driven by advances in artificial intelligence and the utilisation of AIS data. Researchers have been actively working towards enhancing the safety and efficiency of maritime systems, particularly in the context of collision avoidance.

A comprehensive survey by Tu et al. (2018) examines the exploitation of AIS data for intelligent maritime navigation. AIS, which tracks vessel movements through electronic data exchange, offers valuable insights for maritime safety, security, and efficiency. This survey explores various facets of AIS data sources and their applications, including traffic anomaly detection, route estimation, collision prediction, and path planning. It underscores the synergy between data and methodology in marine intelligence research, highlighting the importance of both components.

Although the utilisation of autoencoders in vessel trajectory prediction shows promise based on theoretical merits, some concerns can be raised about their practical application. For instance, the study presented by Murray & Perera (2020) by Murray and Prasad Perera proposes a novel dual linear autoencoder method for predicting vessel trajectories using historical AIS data and unsupervised learning for trajectory clustering and classification. This technique offers several advantages, such as generating multiple possible trajectories and estimating positional uncertainties through latent distribution estimation. However, critics argue that the method may only provide accurate predictions for short time horizons, typically less than 30 minutes, given its reliance on a single initial condition or current position in the trajectory. Additionally, since the algorithm operates based on previously observed states, it might struggle to effectively process new, unseen data. Thus, further investigation into enhancing the adaptability and accuracy of autoencoder-based approaches for longer-term vessel trajectory predictions is warranted.

To address the complexity of maritime traffic patterns and improve Maritime Situational Awareness, Pallotta et al. (2013) have introduced methodologies like Traffic Route Extraction and Anomaly Detection (TREAD). TREAD relies on unsupervised and incremental learning to extract maritime movement patterns from AIS data. It serves as the basis for automatic anomaly detection and trajectory projection, enabling more effective synthesis of relevant behaviours from raw data to support decision-making processes. Trajectory predictions are made by context-based tracking algorithms, using the direction of mean velocity and the series of trajectories taken by other vessels of the same type. However, the trajectory of a fishing vessel is highly anarchic and follows the movements of shoals of fish. It is very difficult to model such behaviour by taking account of positions alone. The statistical model used is based on automatic knowledge discovery. We believe that its ability to predict ship movements away from shipping routes is limited because the context of other ships at the same location is reduced or non-existent.

In the context of increasing concerns about collision accidents in the shipping industry, an advanced ship trajectory prediction model has been proposed (Bao et al., 2022). This model combines a multi-head attention mechanism with bidirectional gate recurrent units (MHA-BiGRU) to enhance precision and real-time capabilities in AIS-based ship trajectory prediction. It effectively retains long-term sequence information, filters and modifies historical data, and models associations between past and future trajectory states, ultimately leading to improved accuracy. But, the approach taken in this model raises certain concerns, particularly regarding the normalisation of longitude and latitude data. While this is a common practice in trajectory prediction models, it can lead to potential inaccuracies. Normalising these coordinates to a 0-1 scale can distort critical spatial relationships, especially near meridians and poles. This simplification might also inadequately represent the real-world physical distance, as a degree's distance varies with latitude. Such distortions are crucial in the shipping industry, where precision in trajectory prediction is paramount for collision avoidance. In 2022, the French Bureau d'enquêtes des évènements de mer (BEAMer) recorded 472 accidents, 68% of which were caused by fishing vessels (BEAMer, 2022).

Urban mobility research has also contributed to the understanding of human transportation modes. TrajectoryNet (Jiang et al., 2017), a neural network architecture for point-based trajectory classification using GPS traces, offers an innovative perspective. It embeds the original feature space into a higher-dimensional representation, enriches the feature space with segment-based information, and employs Maxout activations to enhance the predictive power of Recurrent Neural Networks (RNNs). TrajectoryNet achieves impressive classification accuracy when identifying various transportation modes from GPS traces. Significantly, we were particularly inspired by the application of NLP techniques in this context. The concept of embedding, commonly used in NLP to convert words into vector representations, resonated with us for its potential in our own research. We admired how this approach effectively captures the underlying semantics of continuous features, like speed, in varying contexts. This insight guided us in designing our experiment, where we aimed to similarly interpret and represent the semantics of continuous geospatial features, providing a richer understanding of the data beyond its numerical value.

While the use of Automatic Identification System (AIS) data for predicting vessel trajectories has seen significant advancements, such as the successful deployment of the TrAISformer model by Nguyen & Fablet (2024), however, there are a number of areas for improvement regarding this approach. The TrAISformer model represents an innovative step forward by employing a discretised, multidimensional encoding of AIS data and introducing a distinctive loss function tailored to handle variability and multiple modes within shipping movement data. However, it is important to note that traditional spatial binning techniques (Nguyen et al., 2018) have proven effective for local entity analysis. Nevertheless, these methods face considerable challenges when attempting to scale up for global applications due to resource limitations. Therefore, while the TrAISformer model presents a promising solution, further research is required to assess its efficacy under diverse operational conditions and evaluate its potential impact on computational resources.

A recent study by Spadon et al. (2024) introduces a novel approach to enhance vessel trajectory forecasting through the integration of probabilistic modeling and advanced neural network techniques. The authors made significant contributions to maritime safety through vessel trajectory forecasting. One key innovation is the use of a probabilistic model that leverages historical AIS data to create Route Probability and Destination Probability Matrices. These matrices generate probable routes and destinations, adding valuable features to the vessel trajectory data. This approach significantly enhances the model's decision-making capacity, particularly for complex or curved paths, by reducing mean and median errors. The integration of these probabilistic features allows the model to distill generalized patterns and rules from the data, making it more robust and adaptable to various conditions.

Another notable aspect is the positional-aware attention mechanism, which prioritizes more recent time steps of the input sequence. This methodological focus ensures that recent data points have a greater impact on the learning process, enhancing the precision of the trajectory reconstruction task. By giving higher importance to the latest time steps, the model can capture nuanced temporal patterns that improve its trajectory-fitting capabilities. This approach allows the model to navigate the complexity of maritime navigation more effectively, making it well-suited for real-time applications processing streaming AIS data.

The authors' methodology emphasises the significance of recent data points in the learning process. However, similar to the TrAISformer model (Nguyen & Fablet, 2024), the dataset used in this study is limited to a specific geographical region. Although the objective is not to generalise across all global trajectories, the application of a probabilistic model to enhance learning indicates that the transferability of this model to other regions may require recalculating the probability matrix.

While exploring the literature, we encountered several other methods for ship trajectory prediction (Yang et al., 2022; Li et al., 2024; Sørensen et al., 2022). However, these methods are not autoregressive and do not align with the specific scope and requirements of our study. As a result, they have been excluded from our detailed analysis and comparison.

These various projects represent a selection of research activities that have made significant contributions to the advancement of ship trajectory prediction, the use of AIS data and maritime safety. This research study builds on these foundations and presents a unique approach to GNSS coordinate discretisation and trajectory prediction using transformer architectures.

# 4    Methodology

This section details the methodology described above. The cornerstone of our approach is the transformation of the traditional latitude/longitude coordinate system into a pseudo-octal format using the H3 index. On this basis, we deploy a simplified version of the Mixtral 8x7B model specifically adapted for trajectory prediction. The aim of this adaptation is to make the model accessible for execution on consumer computers while minimising its resource requirements. To assess the accuracy of our predictions, we introduce the use of the Fréchet distance, which serves as a robust measure for evaluating the quality of predictions. The summary is given in the figure 1.

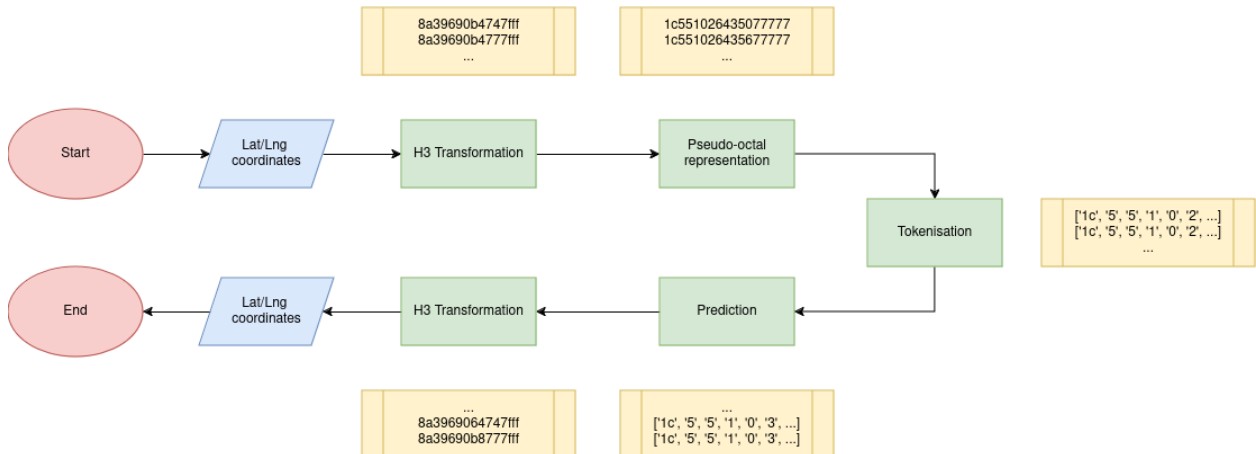

**Figure 1: Flowchart of the method presented throughout the article.** *Green rectangles represent transformation processes, yellow rectangles correspond to examples.*

## 4.1   H3 Index

The methodology section of this research paper delves into crucial decisions that form the basis of our innovative approach in maritime trajectory prediction. In this section, we explore the pivotal choice of utilising the H3 index developed by Uber as the cornerstone of our spatial representation.

Initially, we considered representing each spatial point using a 768 dimensional vector learned through an autoencoder — a model renowned for its effectiveness in conventional contexts. However, we confronted a substantial challenge unique to the realm of geographical coordinates. Even the slightest deviations in these coordinates translated into significant errors in the real-world domain, as previously discussed in the introduction. This limitation prompted us to look for a more robust solution.

The discretisation of a near-continuous space such as the GNSS coordinate system[2], through the use of the H3 index is a rational choice for mitigating inaccuracies in geographical coordinates. By representing spatial data as discrete hexagons, the H3 index ensures that small deviations in coordinates do not translate into significant errors. Each hexagon serves as a well-defined spatial unit, reducing the sensitivity of measurements to minor fluctuations in GNSS data. This contrasts with continuous coordinate systems, where slight variations can lead to substantial discrepancies. Furthermore, the multiple resolution levels of H3 allow for adaptability across different spatial scales, ensuring both accuracy and computational efficiency. This approach not only alleviates the impact of measurement uncertainties but also enhances the robustness of spatial data representation, making it a practical solution for maritime trajectory prediction.

In addition to its robustness, the H3 index offers several advantages that align well with our objectives. H3 represents geographical locations as hexagons, ensuring that any two neighbouring hexagons are always at an

---

[2]Although the GNSS coordinate system is technically discrete, with a smallest measurable unit of 1 mm, we refer to it as "near-continuous" because this level of precision is negligible compared to the scale of decimal degrees. For instance, the first digit in decimal degrees corresponds to a precision of approximately 111 km (ISO 6709:2008/Cor 1:2009).

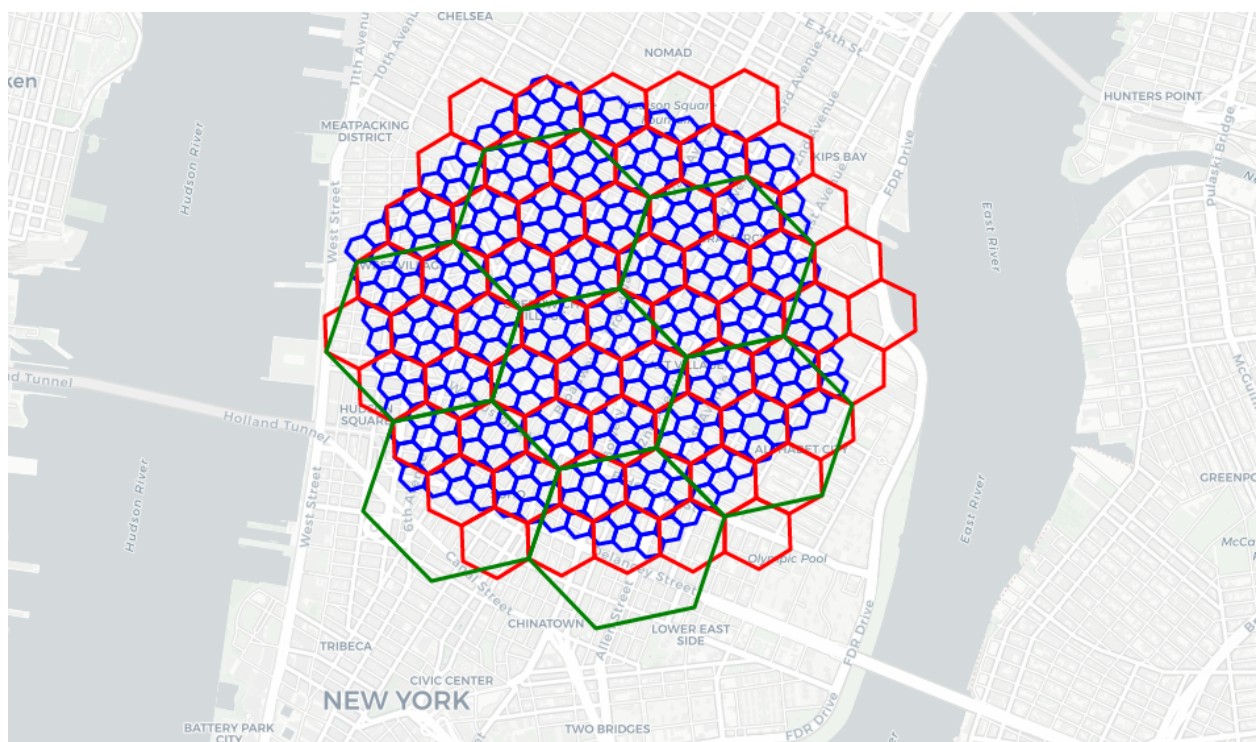

**Figure 2: Illustration of the hexagons of the H3 index at different resolutions over Manhattan centred on the latitude/longitude coordinates 40.73129, -73.99288.** *Resolution 10 is shown in blue, resolution 9 in red and resolution 8 in green.*

absolute distance of 1, unlike the S2 representation system (Google, 2015). The geometric nature of hexagons makes them less deformable at the poles, and H3 provides multiple resolution levels, making it adaptable to various spatial scales. The 64-bit cell identifiers in H3 are highly efficient for handling extensive geographical data, aligning seamlessly with the demands of big data applications. Unlike alternative methods (Sahr et al., 2003), such as Geohash and Hexbin, H3 demonstrates reduced visual distortions when subjected to different map projections, ensuring the preservation of spatial fidelity.

In our research, we opted for a resolution level of 10, which effectively covers an area of approximately 1.5 hectares. To put it in perspective, this resolution corresponds to a square with sides of approximately 122 metres, roughly equivalent to the area of 1.4 football fields. This choice represents a balanced trade-off between the number of cells and computational efficiency, facilitating the practical implementation of our methodology. It's important to note that pursuing greater precision in spatial representation could lead to diminishing returns, as GNSS inaccuracies could result in non-stop coordinate jumps, rendering excessively high resolutions impractical for our purposes.

To facilitate efficient representation and manipulation of H3 cells, we introduced the concept of a pseudo-octal representation. This involved discarding the initial 12 bits (Table 4 in A), considering a fixed resolution mode, and encoding the basic cell as a 2-character hexadecimal string. Each resolution level was then encoded in octal form, ensuring a consistent and concise representation that streamlined subsequent processing steps. Detailed algorithms for this transformation and its inverse can be found in algorithms 1 and 2.

The proposed algorithms for converting H3 cell identifiers to and from a pseudo-octal representation are crucial for optimising the spatial encoding process and addressing limitations in traditional representations. The default hexadecimal representation of the H3 index is inadequate for our use case, as it merely translates the cell identifier without encoding spatial characteristics. This makes it cumbersome for our tokenisation process, as it does not map directly to the hierarchical structure of H3 cells. Likewise, using the 64-bit binary representation would necessitate extremely long sequences for each position, limiting the efficiency of

trajectory predictions. In contrast, the pseudo-octal representation leverages the hierarchical nature of the H3 index, where each subsequent resolution is formed using a 7-cell aperture. By discarding the initial 12 bits and encoding each resolution level in octal form (except the base cell, which remains in hexadecimal), we achieve a compact and efficient representation aligned with the spatial hierarchy of the H3 index. This encoding scheme is particularly valuable for our application in predicting maritime trajectories, where optimising sequence length enhances the model's ability to predict trajectories accurately and efficiently.

---

**Algorithm 1** Converts an H3 cell identifier to a pseudo-octal string representation.

---

**Require:** $cell$ $\triangleright$ a string representing the H3 cell identifier in hexadecimal format
**Ensure:** A pseudo-octal string representation of the given H3 cell identifier
$\quad i \leftarrow h3::str\_to\_int(cell)$
$\quad l \leftarrow []$
$\quad bc \leftarrow format(f >> (52 - 7), 02x)$
$\quad \textbf{for } u \text{ from } 42 \text{ down to } 0 \text{ step } -3 \textbf{ do}$
$\quad\quad digit \leftarrow ((f >> u)\&7) + ORD("0")$
$\quad\quad \text{append } char(digit) \text{ to } l$
$\quad \textbf{end for}$
$\quad \textbf{return } bc + join(l)$ $\triangleright$ Example: $8a39690b4747fff \mapsto 1c551026435077777$

---

**Algorithm 2** Converts a pseudo-octal cell identifier to a hexadecimal string representation.

---

**Require:** $cell$ $\triangleright$ a string representing the H3 cell identifier in pseudo-octal format
**Ensure:** A hexadecimal string representation of the given pseudo-octal cell identifier
$\quad bc \leftarrow cell[:2]$
$\quad value \leftarrow (((1 << 7)|10) << 7)|int(bc, 16)$
$\quad \textbf{for each } c \in cell[2:] \textbf{ do}$
$\quad\quad value \leftarrow (value << 3)|int(c, 8)$
$\quad \textbf{end for}$
$\quad \textbf{return } value$ $\triangleright$ Example: $1c551026435077777 \mapsto 8a39690b4747fff$

---

Tokenisation plays a pivotal role in managing this spatial representation within our text-based model. Each two-character base cell is treated as a single token, with every subsequent resolution level encoded as a distinct token. Since we use resolution 10, all tokens beyond this resolution (levels 11 to 15) are non-informative and set to 7 (an empty octal value). While we could have removed these or tokenised them as a single token to reduce computational complexity, we opted to retain them in our methodology for now. This leaves open the possibility of revisiting the tokenisation in the future, should we choose to adjust the resolutions without modifying the algorithms.

In the following sections, we look at the practical implementation of these choices, showing their impact on improving the prediction of maritime trajectories. Our meticulous approach to spatial representation and tokenisation prepares the ground for the transformative application of Causal Language Modelling (CLM) in maritime trajectory prediction, a promising way to raise the accuracy and efficiency of models in this domain.

## 4.2 Model Architecture

In the pursuit of predicting ship trajectories, the selection of an appropriate model architecture is crucial. This section delves into the rationale behind choosing a causal language model and explores the intricacies of the Mixtral8x7B framework. Causal models, with their auto-regressive nature, are particularly suited for tasks that demand sequential prediction, making them ideal for capturing the temporal and causal structure of the trajectories. This section will detail the specific configuration of our architecture, highlighting its capacity and flexibility in handling the complexities of maritime trajectory prediction.

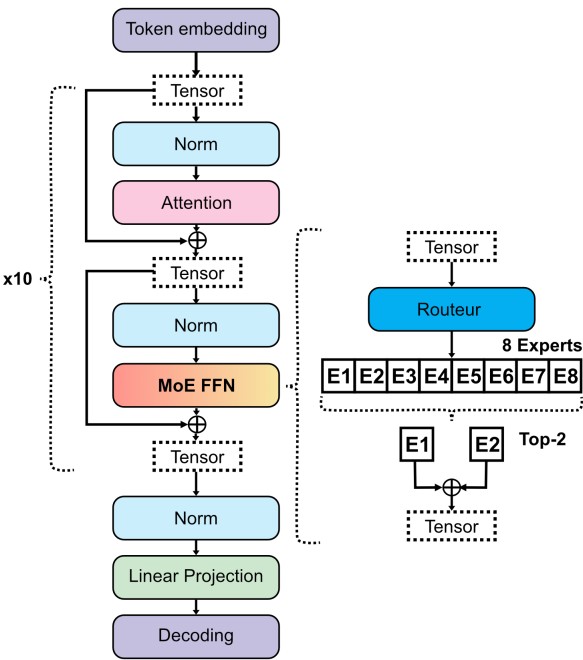

**Figure 3: Model architecture of MixTRAJ**. *We use the same base as Mixtral, with a Mixture of Experts. The only major modification is the vocabulary size (270) as we built our own tokenizer to work accordingly with the pseudo-octal representation. See table 1 for details. Figure from MistralKit (2023).*

### 4.2.1 Rationale for Selecting a Causal Model

The choice of a causal language model, also referred to as an auto-regressive model, is grounded in its inherent ability to predict future tokens based solely on preceding ones (Radford, 2018; Radford et al., 2019; Brown et al., 2020). This auto-regressive framework aligns naturally with many linguistic tasks, as language generation is inherently sequential. In CLM, each word or token is predicted based on all the prior tokens, capturing the temporal and causal structure of language. This makes CLMs particularly effective for tasks where the generation of each token depends on the context provided by the previous ones (unidirectional, left-to-right in our case).

One of the primary strengths of causal models is their ability to handle long-range dependencies within a sequence. Given that language often exhibits dependencies between distant elements, CLMs are well-suited for capturing these relationships over extended sequences. As a result, they can generate text that remains coherent and contextually appropriate over longer spans, which is critical for many downstream applications like our case.

In the context of predicting ship trajectories, the use of a causal language model is particularly justified by the sequential nature of vessel movement, where each GNSS coordinate is dependent on prior positions. By transforming each coordinate into a pseudo-octal representation, we effectively discretise the spatial domain, capturing the hierarchical and spatial characteristics of each location. The auto-regressive nature of the causal language model ensures that each subsequent position is predicted based on the previous sequence of H3 cells, maintaining consistency in the spatial path. This approach is particularly robust in scenarios where long-range dependencies must be captured—such as forecasting future movement over extended distances—while the structured encoding facilitates the model's understanding of spatial continuity.

### 4.2.2 Mixtral8x7B

Our research leverages the powerful architecture provided by the Mixtral8x7B model (Jiang et al., 2024). This model is based upon the Mistral 7B architecture (Jiang et al., 2023) and incorporates a Mixture of Experts approach (Shazeer et al., 2017; Zoph et al., 2022), significantly enhancing its capacity to handle complex tasks. Each expert can focus on different part of the pseudo-octal representation, which improves predictive accuracy by enabling more granular analysis.

The inclusion of a sliding window mechanism further sets it apart, allowing for an attention span of 2560 tokens—a feature particularly beneficial when dealing with lengthy trajectories. In our specific case, this translates to an attention span of over 2 hours, providing the ability to capture intricate patterns in maritime movements. Additionally, the utilisation of Grouped Query Attention not only facilitates faster inference but also reduces cache size requirements, ensuring efficient model operation.

Additionally, Mixtral's integration with the HuggingFace Transformers library (Wolf, 2019) made it an appealing option for development. This compatibility simplifies the model's implementation and ensures that our solution is maintainable and extensible[3].

The complete architecture configuration for our application is defined as follows in the table 1 and can be seen in the figure 3. This comprehensive architecture with 220M parameters is tailored to our maritime trajectory prediction task, offering the capacity and flexibility to handle the complexities of ship movement patterns.

**Table 1:** Mixtral8x7B Model Configuration

| Parameter | Value |
|---|---|
| Vocabulary Size | 270 |
| Hidden Size | 1024 |
| Sliding Window | 256 |
| Number of Hidden Layers | 10 |
| Number of Attention Heads | 16 |
| Number of Key-Value Heads | 8 |
| Number of Experts | 8 |
| Experts per Token | 2 |

### 4.3 Dataset, hardware and training process

The model was trained on a machine equipped with an Intel Core i9-13900K processor, an NVIDIA GeForce RTX 4090 graphics card with 24GB of RAM on a Manjaro 23.1.3 operating system. The model was implemented using the HuggingFace and the PyTorch frameworks.

### 4.3.1 Dataset and Preprocessing

The AIS data utilised in this study were collected over a period of one week in August 2023, within a defined geographical region spanning latitude/longitude coordinates from $(-84.88, -180.00; \ 90.00, 179.65)$ as shown in figure 4. While the collection area covers a wide range, the majority of the recorded vessel trajectories are concentrated in the North Sea, the English Channel, the Atlantic Ocean (along the French, Portuguese, and Spanish coastlines), and the Mediterranean Sea.

As highlighted in the introduction, AIS data are derived from GNSS coordinates, which may suffer from inaccuracies. Common issues encountered included erratic jumps in position, sudden "teleportations" of points, and multiple vessels sharing the same Maritime Mobile Service Identity (MMSI), despite the AIS

---

[3]At the time of the revision of this article, more recent models have been released, and we have not reconducted the experiments with these newer models.

standard requiring MMSIs to be unique. While the protocol itself stipulates unique MMSI assignments, this is not strictly enforced, either technically or by software, leading to potential confusion in trajectory analysis.

To address these challenges, we employed several data cleaning and clustering techniques. The first two issues—erratic points and teleportation—were resolved using the Density-Based Spatial Clustering of Applications with Noise (DBSCAN) algorithm, as described by Ester et al. (1996), with specific optimisations implemented by Wang et al. (2020). We set the DBSCAN epsilon parameter to 10, which filtered out points that did not belong to any meaningful trajectory, as isolated points were considered noise.

For the issue of duplicate MMSIs, a more intricate process was required. The data were first grouped by MMSI, and for each group, we implemented a systematic cleaning procedure. We began by removing duplicate positions and timestamps, ensuring only unique points remained. Additionally, any point where the Speed Over Ground (SOG) was recorded as zero was discarded, as such records are typically associated with stationary vessels.

Following this, the data were sorted chronologically, and a secondary DBSCAN with an epsilon value of 0.1 was applied. This allowed for clustering of spatially related points, with clusters representing coherent vessel trajectories. Any points labelled as noise by the algorithm were excluded. Subsequently, we computed the centroid of each cluster, as well as the average distance from the cluster's points to this centroid. Clusters with an average point-to-centroid distance of less than 15 metres were considered artefacts of GNSS imprecision and were removed, as they likely represented stationary vessels (for example, a docked ship).

At this stage, the remaining clusters represented valid trajectories. To detect instances where multiple vessels might share the same MMSI, we examined the temporal gaps between consecutive points, assigning a new group identifier whenever a time difference exceeding five hours was detected. A unique hash was generated for each point based on the cluster and group number, facilitating the identification of distinct vessels using the same MMSI.

We further refined the trajectories by calculating the Haversine distance between consecutive points and estimating the vessel's actual speed, independent of the recorded SOG. Any points indicating a speed exceeding 100 km/h were flagged as "teleportations"[4], and the entire trajectory containing such points was discarded, based on the actual hash. Consecutive clusters were then iteratively merged by comparing the final point of one cluster with the initial point of the next. If the transition was feasible based on the average speed of the last 10 points (within a $\pm5\%$ tolerance), the two clusters were combined into a single trajectory, with cluster B inheriting the hash and group number of cluster A.

Finally, a consolidated hash was generated for each trajectory, incorporating the MMSI, trajectory number, and individual cluster hash. Any resulting trajectories with fewer than 10 points were excluded from further analysis to maintain data integrity and avoid bias from insufficiently sampled paths. To standardise the temporal resolution, we then applied a downsampling interval of one minute, ensuring consistency across all trajectories.

In addition to the trajectory refinement, the next step involved transforming the AIS points into spatially encoded identifiers using the H3 indexing system. As described earlier in the article, this transformation followed the procedures outlined in algorithms 1 and 2, effectively converting the vessel locations into hexagonal cells for more efficient spatial representation.

Once transformed, the tokenisation process was applied to prepare the data for trajectory forecasting. Each unique trajectory was tokenised, with the resulting sequence split into chunks of size 2560 whenever a trajectory exceeded the context size of the model. Overlapping tokens were preserved, and padding was introduced with an attention mask set to zero. This approach allowed us to handle large datasets without compromising model accuracy. The final dataset comprises 1,731,686 unique trajectories spread across approximately 32 million H3 cells. After tokenisation, it consists of 4 billion tokens.

---

[4]Given that the theoretical maximum speed is 188.904 km/h (meaning that the vessel or helicopter is travelling at or above this speed).

This multi-step approach, incorporating spatial clustering, temporal segmentation, H3 transformation, tokenisation, and trajectory filtering, successfully mitigated issues of duplicate MMSIs and erroneous data points, ensuring a high-quality dataset for subsequent analysis and forecasting.

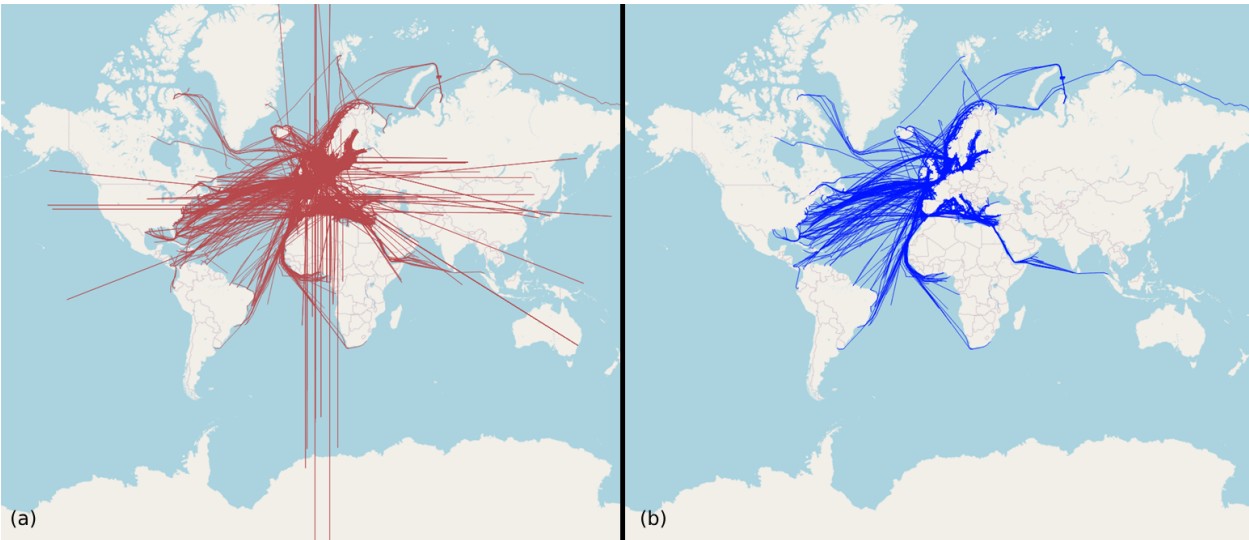

**Figure 4: Comparison of maritime trajectories before and after dataset cleaning.** *(a) Before cleaning, the trajectories exhibit numerous errors, with irregular jumps crossing over land and a high concentration of anomalies covering Europe. These errors indicate a poor capture of the actual ship movements. (b) After applying our cleaning procedure, the trajectories are corrected and closely follow the maritime contours of the continents, removing artefacts and errors that cross landmasses. This process ensures a more accurate representation of the actual maritime routes.*

### 4.3.2 Training and Validation

The training of the model took 175,000 steps using an Adam optimiser with an initial learning rate of 5 times $10^{-4}$ and a gradient accumulation of 8 (see figure 11 in appendix B for the learning curves).

The entire captured dataset has been used for training and validation (respectively 85% and 15% of the dataset). The testing phase was carried out using another dataset that we captured on the fly during 3 days in February 2024 and represents approximately 15,000 cleaned trajectories.

### 4.4 Evaluation metric

To further enhance the evaluation of our trajectory prediction models, we employ the Fréchet distance as a key metric given by the equation 1. This choice is crucial in addressing the limitations of traditional error metrics such as Mean Absolute Error (MAE) or Mean Squared Error (MSE) (Nguyen & Fablet, 2024; Spadon et al., 2024), which compare points in isolation and assume a strict correspondence between them, potentially overlooking the overall shape and progression of the trajectory. The Fréchet distance, by contrast, evaluates the entire trajectory, accounting for both the geometry and the order of points, making it better suited for spatial and temporal analysis. This allows us to capture the continuity and structure of maritime movements more effectively, providing a more comprehensive assessment of model performance as shown in figure 5. We implemented the algorithm using the Haversine function, a method commonly employed for calculating distances between points on the Earth's surface due to its accuracy in spherical geometry.

Consider two curves $A$ and $B$ in the metric space $S$. The Fréchet distance between these curves is the infimum of the maximum distances between points on $A$ and $B$, mapped through reparametrisations $\alpha$ and $\beta$ of the interval $[0, 1]$. This distance measures the dissimilarity between $A$ and $B$ by evaluating the maximal spatial discrepancy across all possible point correspondences established by $\alpha$ and $\beta$. The metric $d$ in $S$ is the Haversine distance, as delineated in equation 2.

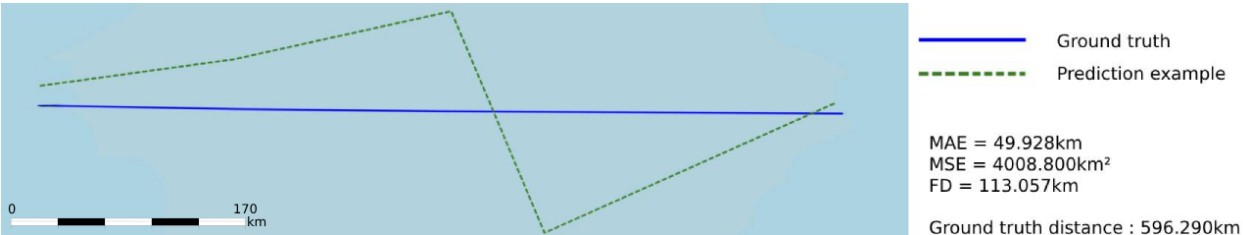

**Figure 5: Comparison of the Fréchet distance versus Mean Absolute Error (MAE) and Mean Squared Error (MSE).** *Over a distance of 596 km, the predicted trajectory deviates significantly from the ground truth, making such a prediction unacceptable. The MAE gives the lowest error, but this value is misleading as it does not capture the true nature of the deviation, being too low due to its point-by-point comparison. In contrast, the Fréchet distance accounts for the overall shape of the trajectories, providing a higher and more realistic measure of error.*

$$F(A, B) = \inf_{\alpha, \beta} \max_{t \in [0,1]} \left\{ d\Big( A\big(\alpha(t)\big), B\big(\beta(t)\big) \Big) \right\}; \tag{1}$$

where $d$ is the distance function of $S$ defined in 2.

$$d \colon \begin{cases} L_1 = \frac{\varphi_2 - \varphi_1}{2} \\ L_2 = \frac{\lambda_2 - \lambda_1}{2} \\ \varphi_1, \varphi_2, \lambda_1, \lambda_2 \longmapsto 2r \arcsin\left( \sqrt{\sin^2\left(L_1\right) + \cos(\varphi_1)\cos(\varphi_2)\sin^2\left(L_2\right)} \right) \\ [-90; 90] \times [-90; 90] \times [-180; 180] \times [-180; 180] \longrightarrow \mathbb{R}_+ \end{cases} \tag{2}$$

where $r$ is the Earth's radius, $\varphi_1$ and $\varphi_2$ are the latitudes,

$\lambda_1$ and $\lambda_2$ are the longitudes.

## 4.5 Qualitative and Quantitative Evaluation of Trajectory Predictions

To the best of our knowledge, there is currently no method capable of predicting ship trajectories globally and over long distances without being tailored to specific geographic regions. Most existing approaches are limited to particular areas, restricting their applicability on a broader scale. In contrast, we sought to develop a solution that is not constrained by these regional limitations.

For the qualitative evaluation on our test dataset, we have chosen to compare our approach with Kalman filters (Kalman, 1960). This widely used method is known for its adaptability in trajectory prediction, yet it still requires regional adjustments, making it an apt baseline to highlight the advantages of our model. In our implementation, only GPS positions—longitude (*lon*) and latitude (*lat*)—are used to predict trajectories. Each prediction is represented by a matrix $G \in \mathbb{R}^{\tau \times 2}$, where $\tau$ is the length of the prediction. To align with our approach, which relies solely on positional data, we restricted the state in the Kalman filter to include only four values: longitude (*lon*), latitude (*lat*), and their respective velocities ($v_{lon}$ and $v_{lat}$). Although Extended Kalman filters can incorporate more complex data such as sea current patterns, we opted to reduce dependency on external data by limiting the parameters of both our model and the Kalman filter. Since the Kalman filter predicts a single state and relies on real observations in its update phase, and real observations are not yet available in our maritime case, we modified the Kalman filter to operate in an auto-regressive mode over $\tau$ steps. In this setup, each predicted state becomes the subsequent input in the prediction cycle, as expressed in equation 3.

$$\begin{aligned} \hat{x}_{k|k-1} &= F_k x_{k-1} \\ P_{k|k-1} &= F_k P_{k-1} F_k^T + Q_k \end{aligned} \tag{3}$$

The state is a vector $x = (lat, lon, v_{lat}, v_{lon})$. The transition matrix $F_k \in \mathbb{R}^{4 \times 4}$ is as follows considering $\Delta_t$ the time between two points, in our case 60 seconds:

$$F_k = \begin{pmatrix} 1 & 0 & \Delta_t & 0 \\ 0 & 1 & 0 & \Delta_t \\ 0 & 0 & 1 & 0 \\ 0 & 0 & 0 & 1 \end{pmatrix}$$

We define $Q_k$ to be the process noise covariance matrix as an identity matrix $\mathbb{R}^{4 \times 4}$ multiplied by a factor of $1 \times 10^{-3}$ and $P_{k|k-1}$ to be the state covariance matrix as an identity matrix $\mathbb{R}^{4 \times 4}$.

Additionally, we conducted a quantitative comparison with TrAISformer and Seq2Seq Sutskever et al. (2014), introduced in earlier sections as a state-of-the-art reference. To ensure consistency, we reproduced the TrAISformer experiment ourselves using the code provided by the original authors. However, we modified the evaluation metric to use the Fréchet distance, as discussed previously, and used TrAISformer's evaluation dataset to measure the performance of our model, MixTRAJ, both with and without fine-tuning.

## 5 Results and discussions

### 5.1 Metrics analysis

Our analysis of the predictive accuracy of ship trajectory models shown in the table 2, centred on the Fréchet distance as a key metric, reveals nuanced insights into the model's performance in real-world navigation scenarios. By applying a stringent selection criterion, we isolated trajectories where the model's predictions were not only completed but also exceeded a prediction length of $\eta$ minutes, as observed in our prior analysis and where $\eta = med(\mathcal{X}) - 5$, with $med(\mathcal{X})$ the median of the series of prediction minutes $\mathcal{X}$. This filtration yielded a dataset conducive to a focused analysis on the effectiveness of our predictive model under conditions of substantial navigational complexity.

The violin plots in figure 6 depict the distribution of two metrics used to evaluate the performance of our model in predicting ship trajectories: the prediction distance and the Fréchet distance. Sub-figure **(a)** shows the distribution of the prediction distance (in metres), calculated as the straight-line distance between the last point of the context and the last point of the predicted trajectory. With a mean of approximately 29,790 metres and a density peak around 32,401 metres, this value represents the average distance covered in 90 minutes of prediction. This metric helps to contextualise the model's output by quantifying the length of trajectory being predicted over time, as our model outputs tokens only.

In sub-figure **(b)**, the Fréchet distance is used to compare the predicted trajectory with the actual one. Fréchet distance measures the similarity between the full predicted and actual paths, capturing the accuracy of the model over the entire trajectory rather than just at the endpoints. This distance is calculated using the closest ground-truth point to the centroid formed by the $N$ predictions in a given context. The mean Fréchet distance is 2,806 metres, with a median of 1,938 metres and a density peak at 1,061 metres. The proximity of the density peaks in both sub-figures suggests that the model performs consistently, with a typical prediction error of around 1 kilometre for an overall predicted distance of 32 kilometres. This demonstrates that the model effectively captures the global patterns of ship movement under most conditions (see appendix C and D).

The long tails in both plots highlight instances where the model's predictions deviate more significantly from the actual trajectories. Such deviations could arise from unpredictable environmental factors (like sudden weather changes), complex maritime traffic conditions, or unique navigational challenges not fully encapsulated by the model's parameters such as the trajectories of fishing boats, which are difficult to predict by taking only GNSS positions into account. These cases underscore the challenges of modelling the intricacies of real-world navigation, where unforeseen variables can impact a ship's course.

The general effectiveness of the model in a wide range of conditions is a strong proof of its utility in maritime operations, offering a robust tool for route planning, risk assessment, and decision-making support.

Mean, median and density peak for a 60-minute contextual window and a 90-minute prediction.

**Figure 6: Violin plot of the prediction distance**. *Without outliers, highlighting the mean (green star, indicated by the green arrow head), median, and density peak for a context length of 60 minutes and a prediction length of 90 minutes. See figures 14, 15 and 16 in E for the full analysis.*

However, the instances of significant prediction error—reflected in the higher Fréchet distances—signal the need for caution. They suggest that reliance on the model should be balanced with situational awareness and potentially supplemented with real-time data and human expertise, especially in navigating complex or unfamiliar waters. By reducing instances of high Fréchet distances, we would improve the model's reliability across a wider array of navigational scenarios.

**Table 2:** Relative deviation between predictions and ground truth for mean, median and peak density. The context is expressed in minutes, while the prediction is expressed in the number of tokens generated. It can be seen that for a doubled distance, the relative error is approximately doubled, which suggests that the model has generalised sufficiently to produce a linear error.

| Context \ Prediction | 2560 | 5120 (x2) | 7680 (x3) |
|---|---|---|---|
| **Mean** | | | |
| 30 (540 tokens) | 11.46% | 18.30% (x1.60) | 22.89% (x2.00) |
| 60 (1080 tokens) | 9.42% | 18.84% (x2.00) | 25.61% (x2.71) |
| 100 (1800 tokens) | 6.97% | 14.78% (x2.12) | 23.33% (x3.34) |
| **Median** | | | |
| 30 (540 tokens) | 7.80% | 13.23% (x1.70) | 17.54% (x2.24) |
| 60 (1080 tokens) | 6.31% | 13.67% (x2.16) | 19.46% (x3.08) |
| 100 (1800 tokens) | 4.88% | 11.33% (x2.32) | 17.82% (x3.65) |
| **Density Peak** | | | |
| 30 (540 tokens) | 4.51% | 5.95% (x1.32) | 10.33% (x2.29) |
| 60 (1080 tokens) | 3.27% | 6.51% (x1.99) | 7.98% (x2.44) |
| 100 (1800 tokens) | 2.67% | 6.86% (x2.57) | 8.55% (x3.20) |

The figure 7 highlights that the prediction error generally increases with the prediction distance, as shown by the comparison between (a) and (b). It is generally observed that the prediction errors are greater as

**Table 3:** Comparison of trajectory prediction methods using public AIS data from TrAISformer. The evaluation metric used is the Fréchet distance (in meters). The Seq2Seq models (LSTM and GRU) share the same architecture and were trained on TrAISformer data. TrAISformer was implemented using the source code provided by the original authors.

MixTRAJ models were evaluated in three different scenarios:
- **Few-Shot**: Trained on a private dataset, where 2% of the geographical area overlaps with the TrAISformer dataset for a duration of one week, and evaluated on the TrAISformer dataset.
- **Fine-Tuned**: Trained directly on the public TrAISformer dataset, and evaluated on the TrAISformer dataset.
- **Zero-Shot**: Evaluated on a private dataset from the Singapore Strait, without prior exposure to this region.

| Models | Parameters | FLOPS (G) | MACs (G) | Error (2h) |
|---|---|---|---|---|
| Seq2Seq (LSTM - ours) (Sutskever et al., 2014) | 967k | 0.2 | $9 \times 10^{-5}$ | 15033.4 |
| Seq2Seq (GRU - ours) (Sutskever et al., 2014) | 730k | 0.1 | $9 \times 10^{-5}$ | 6364.0 |
| TrAISformer (Nguyen & Fablet, 2024) | 57.5M | 13.7 | 6.8 | 6945.2 |
| MixTRAJ with finetuning (ours) | 221M | 4044.1 | 2018.8 | **2528.4** |
| MixTRAJ (Few-Shot task, ours) | 221M | 4044.1 | 2018.8 | 5407.6 |
| MixTRAJ (Zero-Shot task, ours) | 221M | 4044.1 | 2018.8 | 6198.7 |

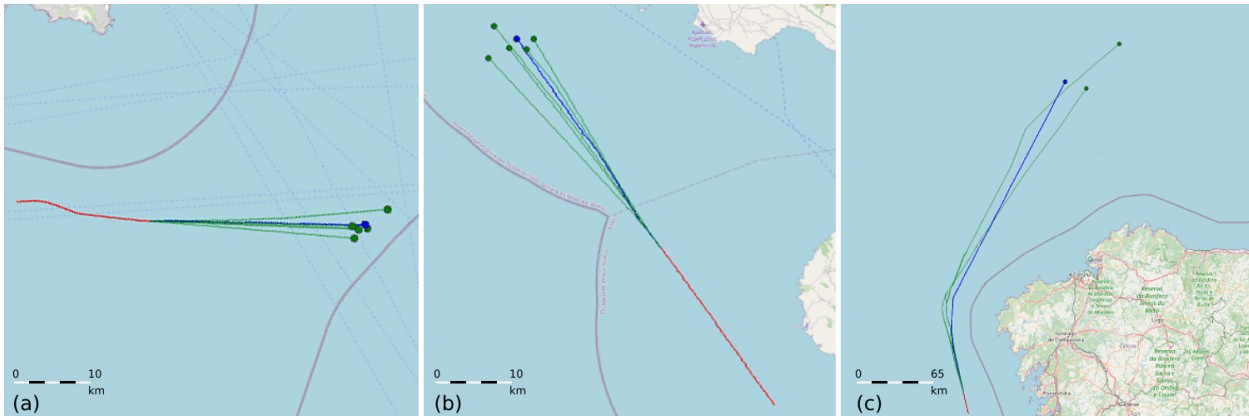

**Figure 7: Path prediction performance comparison.** *This graph compares trajectory prediction performance for different configurations of context windows and prediction distances. (a) 60 minutes of context with a prediction at 90 minutes, (b) 100 minutes of context with a prediction at 90 minutes and (c) 60 minutes of context with a prediction at 20 hours. The colours represent the context window (red), the trajectories predicted by the model (green) and the actual trajectory (blue). We can see zigzags between the points due to the geometric arrangement of the hexagons. (a) and (b) are to the same scale, (c) is zoomed out.*

the context length is reduced, and the prediction length is increased. On the other hand, we can see that, despite certain hallucinations which we will discuss later, the model seems to have generalised the nature of a trajectory, as shown in image (c) of figure 17 in appendix F.

We compare our model to the Kalman filter on the same dataset. However, we did not apply the H3 transformation for the Kalman pass, as the filter is then less sensitive to trajectory zigzags. We changed the colour of the predictions to yellow for the Kalman filter in order to differentiate them from the green predictions of our model.

The figure 8 depicts the results of the Kalman filter. We notice that the Kalman filter appears to draw a straight vector from the last known position. As we know, our auto-regressive Kalman filter does not take into account the history of the ship and a Kalman filter is not designed to know the trends of other

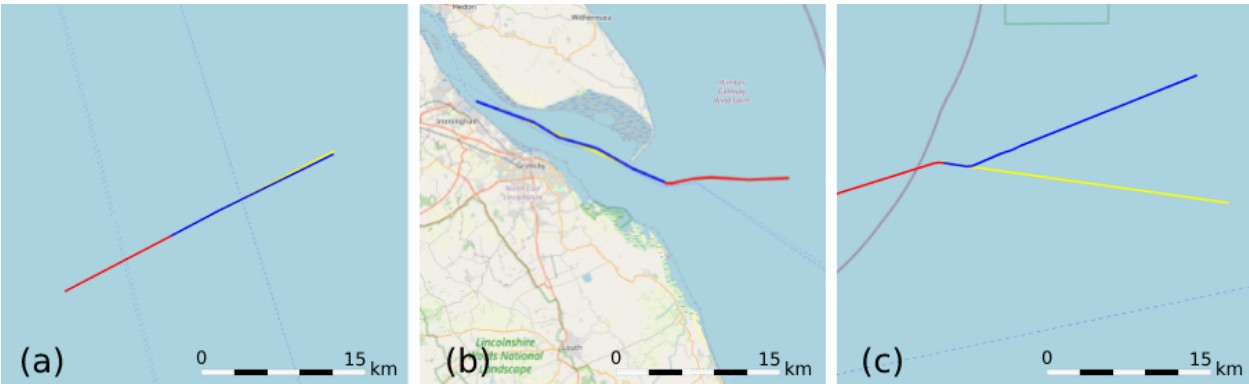

**Figure 8: Context (red), ground-truth (blue) and prediction (yellow) using a Kalman filter.** *The figure shows that the Kalman filters, in our use case, predict linear trajectories because they are auto-regressive, and they are not very sophisticated (e.g. no integration of headings and marine drift models). This seems to work very well when a ship is going straight along its route (a, b), but not so well when there are sharp bends or changes of direction (c).*

ships. We are trying to demonstrate here that the Kalman filter is not a totally reliable indicator for a ship because it has no knowledge of the ship's past, the ship's current trend or the shipping trend of other ships. However, we can qualify this by considering that a Kalman filter can be good and highly sufficient on a defined maritime route (for example for cargo ships and container ships) because these ships have straight trajectories.

In contrast, as shown in Table 3, MixTRAJ demonstrates a substantial enhancement in predictive accuracy, outperforming the baseline TrAISformer model by 28.4% without fine-tuning and by an impressive 174.7% with fine-tuning on TrAISformer's evaluation dataset. This result highlights that, even without fine-tuning on TrAISformer's data and despite the limited representation of this specific region in our initial dataset, our baseline model performs better, suggesting it has successfully generalised the intrinsic nature of trajectory patterns. Notably, we opted to use the median error for model evaluation, whereas the initial TrAISformer implementation used the minimum error; we adapted this to ensure a more robust assessment.

Additionally, we evaluated our model through a Zero-Shot task in a region unseen before (table 3). This region is a zoomed-in view of the Singapore Strait, from $(15.44, 92.91)$ to $(-9.57, 128.03)$ in latitude/longitude coordinates. We observed that our model performs relatively well in this new context. This Zero-Shot evaluation underscores the model's ability to generalize to new and unseen regions while also indicating the importance of fine-tuning for achieving the highest accuracy. It remains the most effective approach to adapt to the specific characteristics of a region that influence trajectory patterns.

### 5.2 Trajectory density

In our study, we aim to assess the veracity of ship trajectories by generating a set of thirty predictive paths. The primary objective is to evaluate the accuracy of these trajectories by examining the proximity of the centroid of a polygon, formed by the endpoints of the generated trajectories, to the actual ground truth position of the ship. This method provides an estimate of the reliability of the predictions made by our model.

We have observed two distinct scenarios in the outcomes of our analysis. In the first scenario depicted in figure 9, a significant majority of the trajectories coincide, indicating a high likelihood that these paths accurately reflect the ship's movement. This consensus among the trajectories suggests a precise and reliable prediction. In the second scenario, the generated trajectories vary widely, resulting in what could be described as chaos (figure 10, left). However, this dispersion does not necessarily indicate a flaw in the model but rather a multitude of possible paths the ship could take. In such cases, further analysis and additional context are required to ascertain the exact trajectory of the ship.

Lastly, it is crucial to highlight that some trajectories generated by our model are physically unfeasible for a ship to execute. We have coined the term "hallucinations" to describe these unachievable trajectories, a concept further elaborated in the following section of our paper. This distinction aids in refining the accuracy of our model by eliminating non-viable predictions and focusing on plausible trajectories.

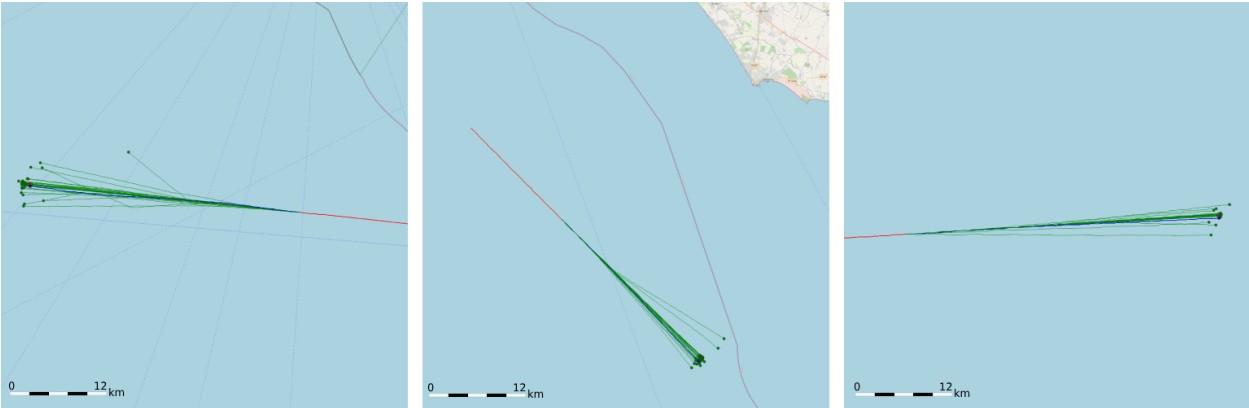

**Figure 9: Illustration of the trajectory consensus phenomenon.** *This figure shows three examples of sets of trajectories where between 85 and 95% of the predicted trajectories align closely, representing a high degree of agreement between the model's predictions. Each set demonstrates the model's ability to accurately predict ship motion, highlighting the reliability of consensus trajectories in reflecting the actual navigation path.*

## 5.3 Hallucination

In the development of this predictive model for ship trajectory forecasting, an interesting phenomenon that we called "hallucination" has been observed. This term is used to describe generated trajectories by the model that are not feasible for a ship to execute in reality. Examples of such unrealistic trajectories include abrupt U-turns, tight turns, and improbable GPS jumps, which are highly unlikely in our context. These anomalies can be visually identified in the model's output, as showcased in the subsequent figure 10.

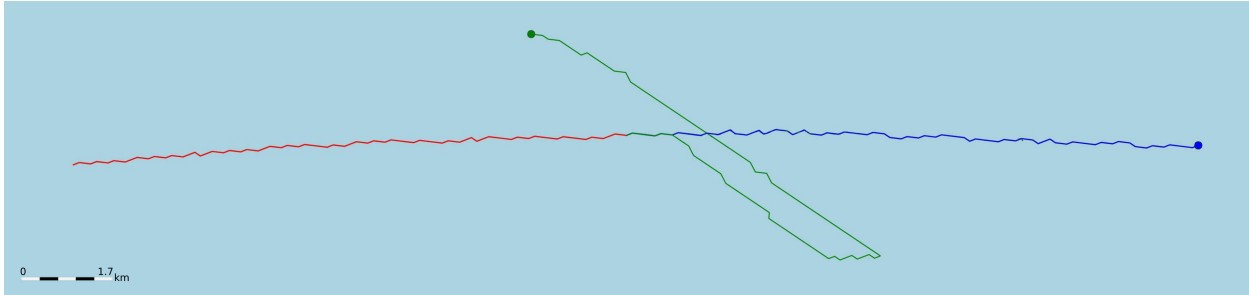

**Figure 10: An example of the phenomena we have called "hallucinations".** *A red line represents the context used to generate the green trajectory. A blue line represents the ground truth. See more example in the figures 18 and 19 in appendix G*

One of the key reasons behind the occurrence of these hallucinations is linked to the nature of the Causal Language Model we used and its auto-regressive architecture. In an auto-regressive framework, each prediction is based solely on the previous tokens without any direct consideration of future points, making it prone to accumulating small errors over time. As these errors compound, especially in longer sequences, they can lead to unrealistic outputs such as abrupt turns or improbable position jumps.

To address and potentially mitigate the generation of hallucinatory trajectories, future research could explore the application of reinforcement learning solutions. By incorporating concepts of correct versus incorrect

trajectories, techniques such as Reinforcement Learning from Human Feedback (RLHF) (Schulman et al., 2015; 2017; MacGlashan et al., 2017; Stiennon et al., 2020; Ouyang et al., 2022), Reinforcement Learning with Artificial Intelligence Feedback (RLAIF) (Lee et al., 2023), or Direct Preference Optimization (DPO) (Rafailov et al., 2023) might offer promising avenues for improvement. Additionally, integrating a positioning vector token within a pseudo-octal representation of the cell could enhance the spatial structure understanding of the model, promoting more realistic trajectory predictions.

This exploration into advanced learning methodologies and structural enhancements remains a prospective area of research. Such investigations could significantly refine the predictive accuracy and reliability of ship trajectory models, ensuring that hallucinations are effectively minimized or altogether eliminated in future iterations.

## 6 Conclusion

In conclusion, our foray into the H3 index system has significantly enhanced our approach to spatial data representation, particularly in the context of maritime trajectory prediction. It is important to clarify that our model is designed for manual user actions in GIS software and is not intended for real-time streaming applications. The ability of the H3 index to reduce visual distortions across different map projections and to maintain spatial fidelity has been crucial in our work.

The introduction of a pseudo-octal representation for the H3 cells marked a pivotal point in our research. This innovative approach, involving the discarding of initial bits and encoding cells in a more compact format, significantly streamlined our data processing. The algorithms we developed for this transformation and its inverse showcase our commitment to enhancing the efficiency of spatial data manipulation.

Looking at the practical implementation of these choices, it becomes clear that they have impacted the prediction of maritime trajectories. Our methodical approach to spatial representation and tokenisation has laid the groundwork for the application of Causal Language Modelling (CLM) in this domain. The integration of CLM promises to improve maritime trajectory prediction, enhancing both the accuracy and efficiency of predictive models.

Although the Mixtral8x7B model architecture is very effective at capturing the complex patterns of maritime trajectories, we recognise the potential benefits of Fourier transforms. Recent advances, particularly in FNet (Lee-Thorp et al., 2021), have demonstrated the usefulness of Fourier transforms in sequence modelling tasks. We believe that the application of Fourier transforms could further improve our model's ability to recognise additional patterns and nuances in ship trajectories, which could lead to even more accurate predictions. This line of exploration represents an intriguing future direction in our pursuit of improving the accuracy of maritime trajectory prediction.

This research not only contributes to the field of geospatial data analysis but also opens new pathways for future exploration, particularly in the application with advanced modelling techniques like CLM. We anticipate that our findings and methodologies will pave the way for more sophisticated and accurate applications in maritime trajectory prediction and beyond. Additionally, this research is geared towards manual GIS workflows, highlighting its distinction from real-time applications.

## Data Availability

Due to the sensitive nature of the data, which was collected using a proprietary system of the French Navy, we are unable to share the raw dataset. However, we have provided a detailed description of the data preparation procedure, ensuring that others can replicate the cleaning and tokenisation steps. If needed, we can also provide the code used for data cleaning. Moreover, the data can be recaptured dynamically, as we did during the testing phase, or exported from publicly available paid-platforms such as MarineTraffic and VesselFinder, allowing for similar experiments to be conducted with alternative datasets. For free data, refer to AISHub (https://www.aishub.net/), Norwegian coastguard's server (https://www.kystverket.no/en/navigation-and-monitoring/ais/access-to-ais-data/) or the data history of the Danish Maritime Authority (https://web.ais.dk/aisdata/).

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

# A   H3 Index Bit Layout

**Table 4:** H3 Index Bit Layout

| Start | Bits | Type |
|-------|------|------|
| 0 | 1 | Reserved |
| 1 | 4 | Index Mode |
| 5 | 3 | Mode-Dependent |
| 8 | 4 | Resolution |
| 12 | 7 | Base Cell |
| 19 | 3 | Res 1 digit |
| 22 | 3 | Res 2 digit |
| 25 | 3 | Res 3 digit |
| 28 | 3 | Res 4 digit |
| 31 | 3 | Res 5 digit |
| 34 | 3 | Res 6 digit |
| 37 | 3 | Res 7 digit |
| 40 | 3 | Res 8 digit |
| 43 | 3 | Res 9 digit |
| 46 | 3 | Res 10 digit |
| 49 | 3 | Res 11 digit |
| 52 | 3 | Res 12 digit |
| 55 | 3 | Res 13 digit |
| 58 | 3 | Res 14 digit |
| 61 | 3 | Res 15 digit |

## B    Learning curves

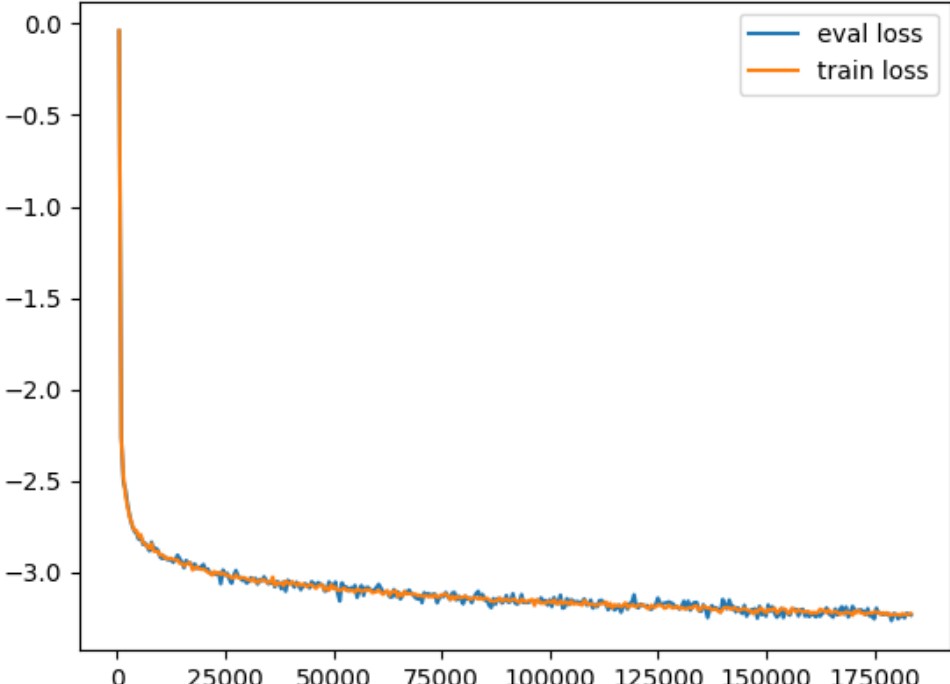

**Figure 11: Unified training and evaluation loss curves.** *It shows the loss value compared to the number of steps in the training process. Convergence of training (orange) and evaluation (blue) loss curves indicates effective model generalisation without overfitting.*

## C    Density plot of Fréchet distance versus prediction distance

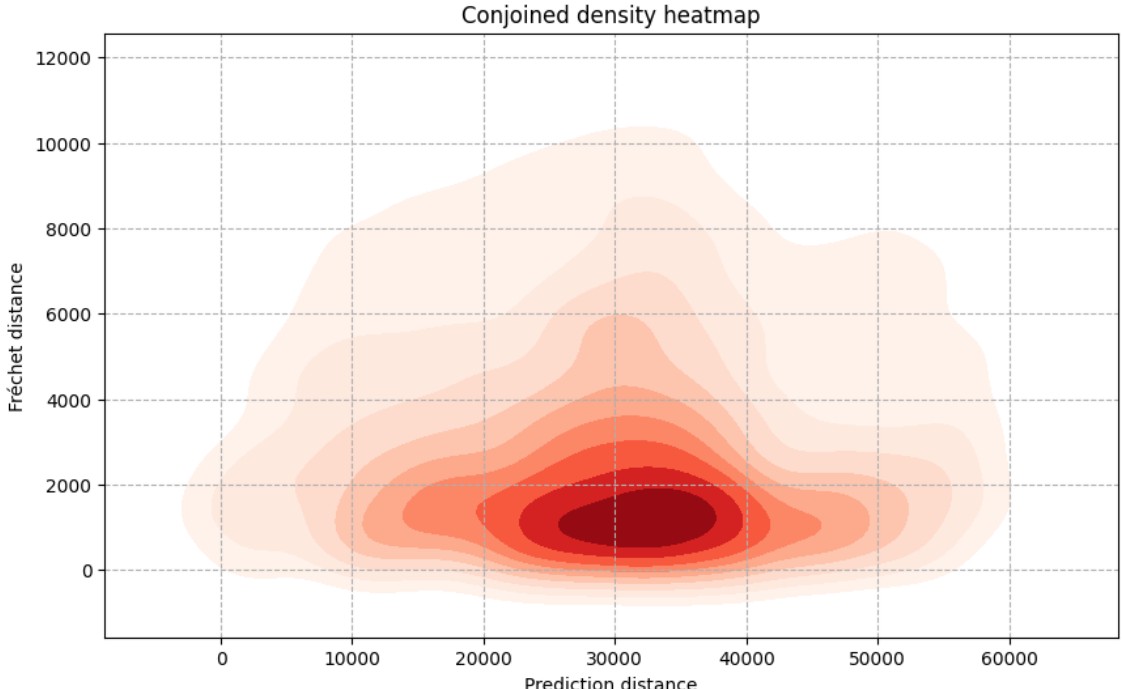

**Figure 12: Graph showing the density of the Fréchet distance in metres compared with the prediction distance in metres for 90-minute predictions with 60 minutes of context.** *We can see that the errors measured by the Fréchet distance are minimal for prediction lengths of around 30km. Prediction distances that are too short or too long can be interpreted as teleportations or hallucinations.*

## D    Example of a river prediction

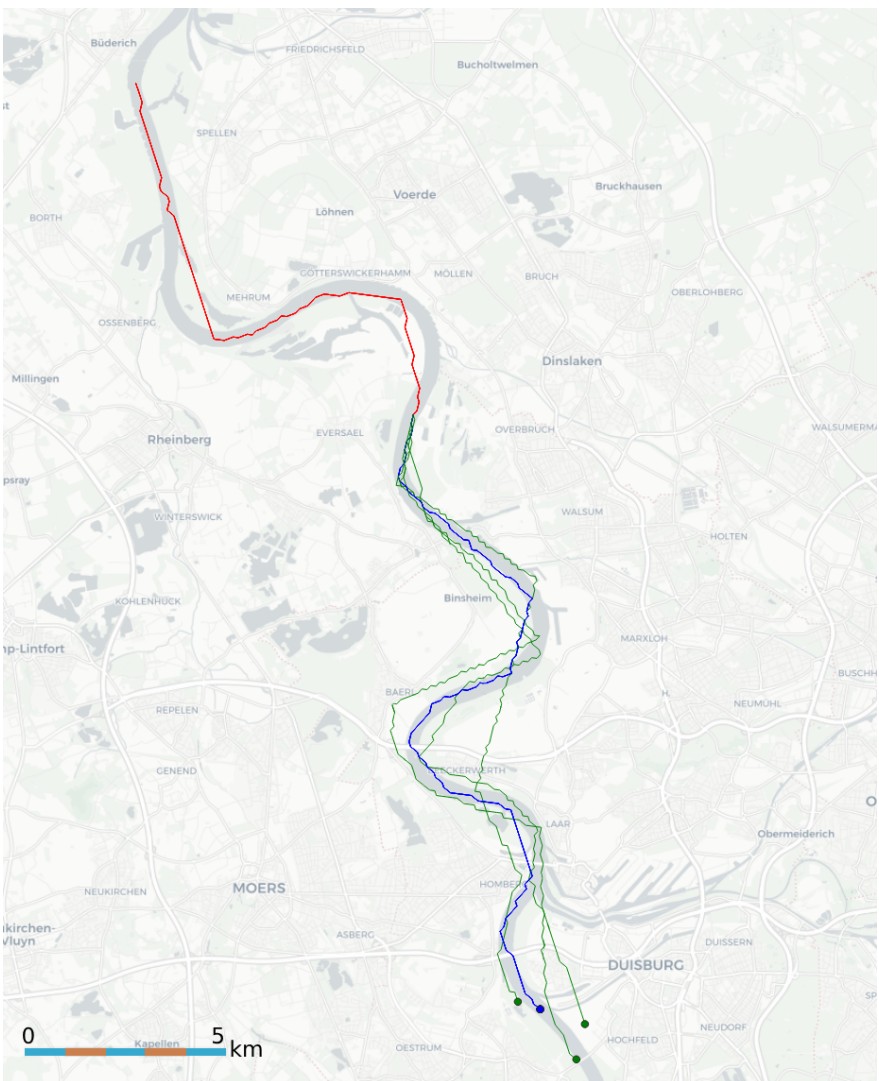

**Figure 13: Illustration of a prediction of a ship's trajectory in a river.** *The context is represented by a 60 minutes long red line, the 155 minutes long green lines represent predictions. The ground truth is shown in blue. It is interesting to note that accuracy in these areas is lower, notably due to the inaccuracy of GNSS.*

# E Distribution of predictions

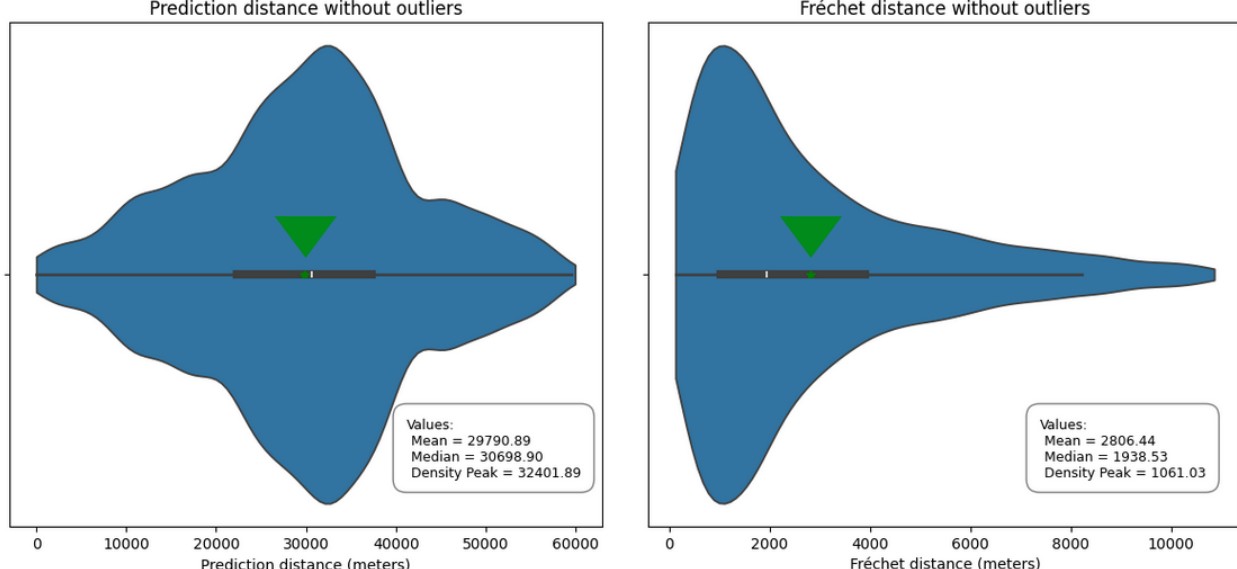

**Figure 14:** *Violin plot of the prediction distance without outliers, highlighting the mean (green star), median, and density peak for a context length of 60 minutes and a prediction length of 90 minutes.*

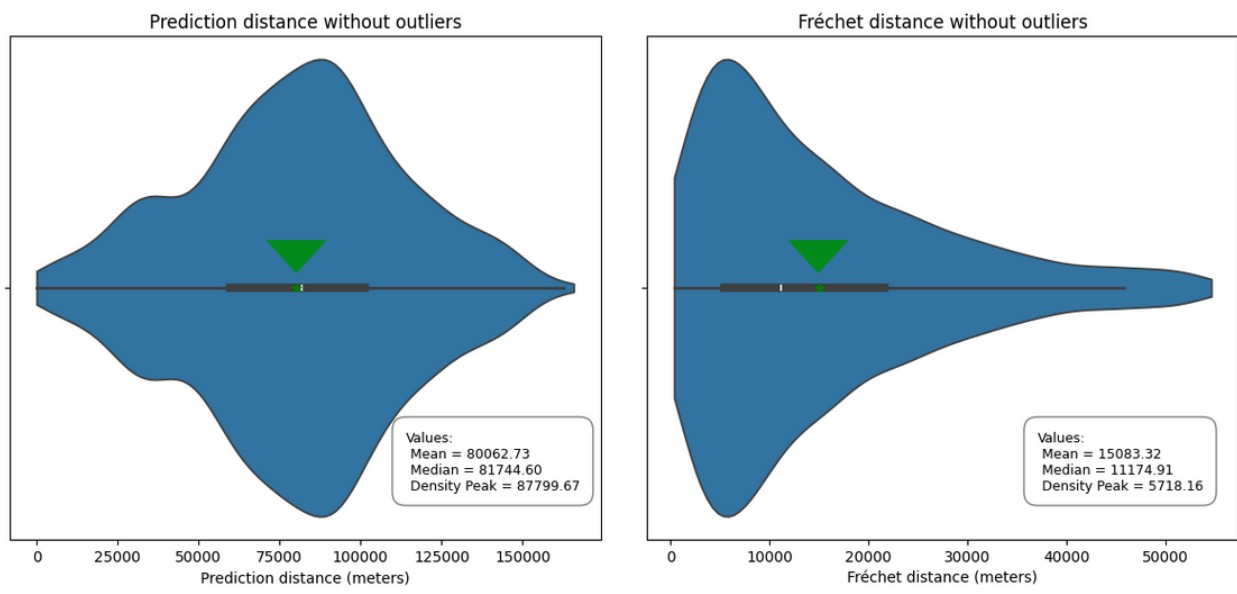

**Figure 15:** *Violin plot of the prediction distance without outliers, highlighting the mean (green star), median, and density peak for a context length of 60 minutes and a prediction length of 240 minutes.*

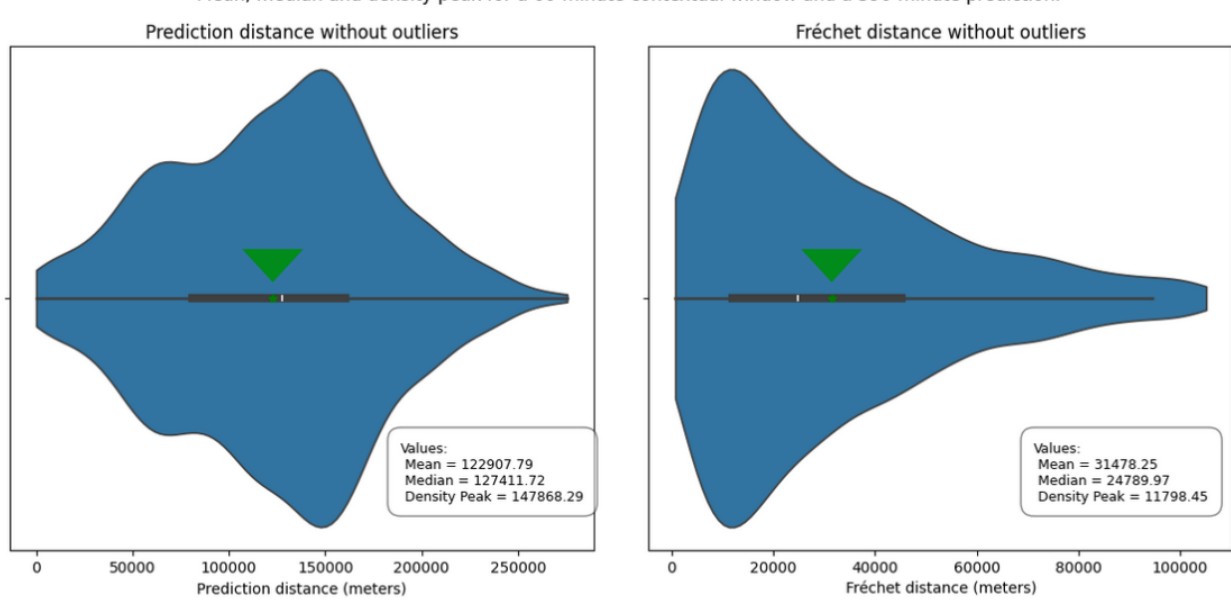

**Figure 16:** *Violin plot of the prediction distance without outliers, highlighting the mean (green star), median, and density peak for a context length of 60 minutes and a prediction length of 390 minutes.*

## F    Multiple trajectories grid

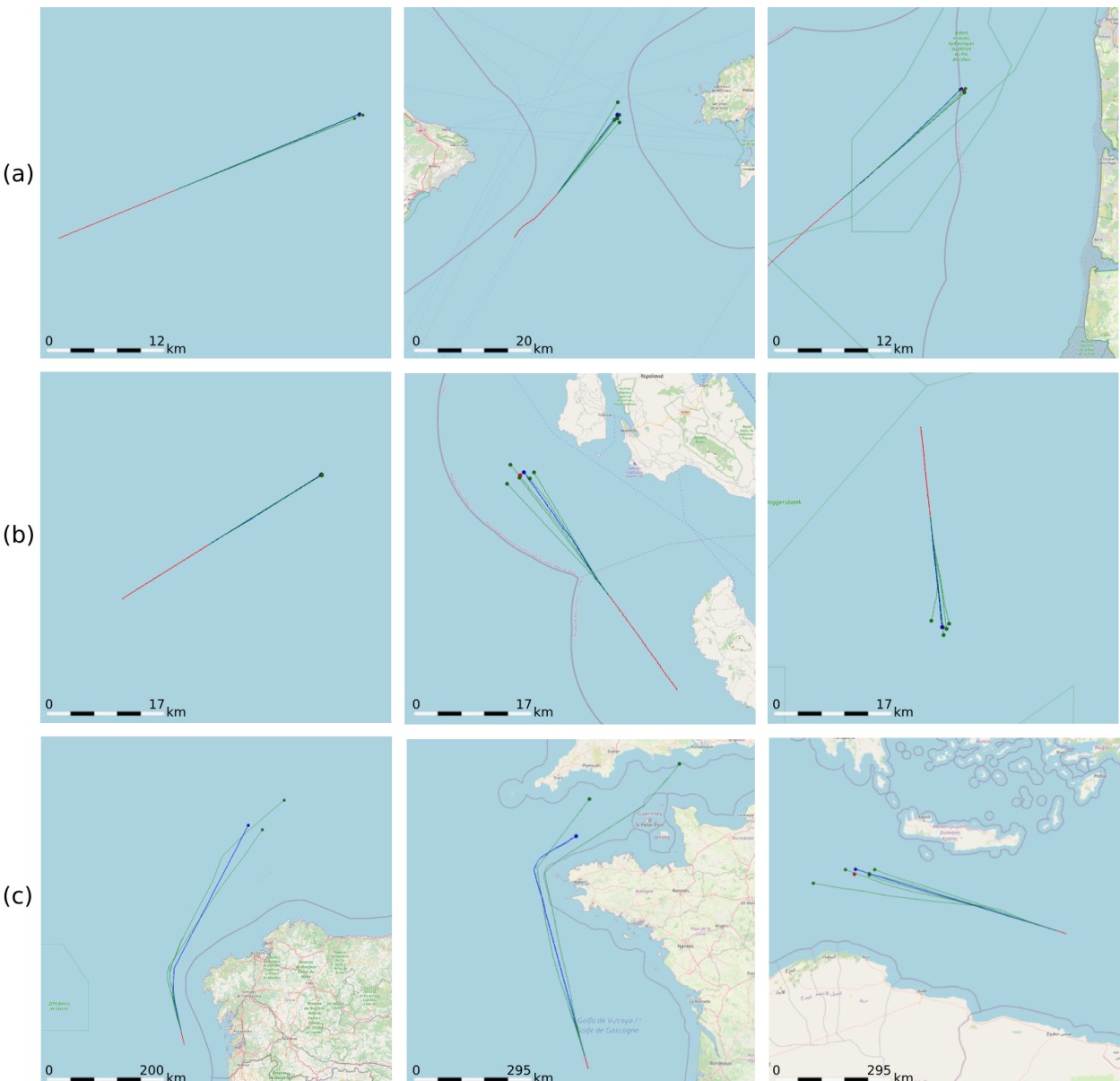

Figure 17: Zoom out path prediction performance comparison of figure 7.

## G    Hallucinations

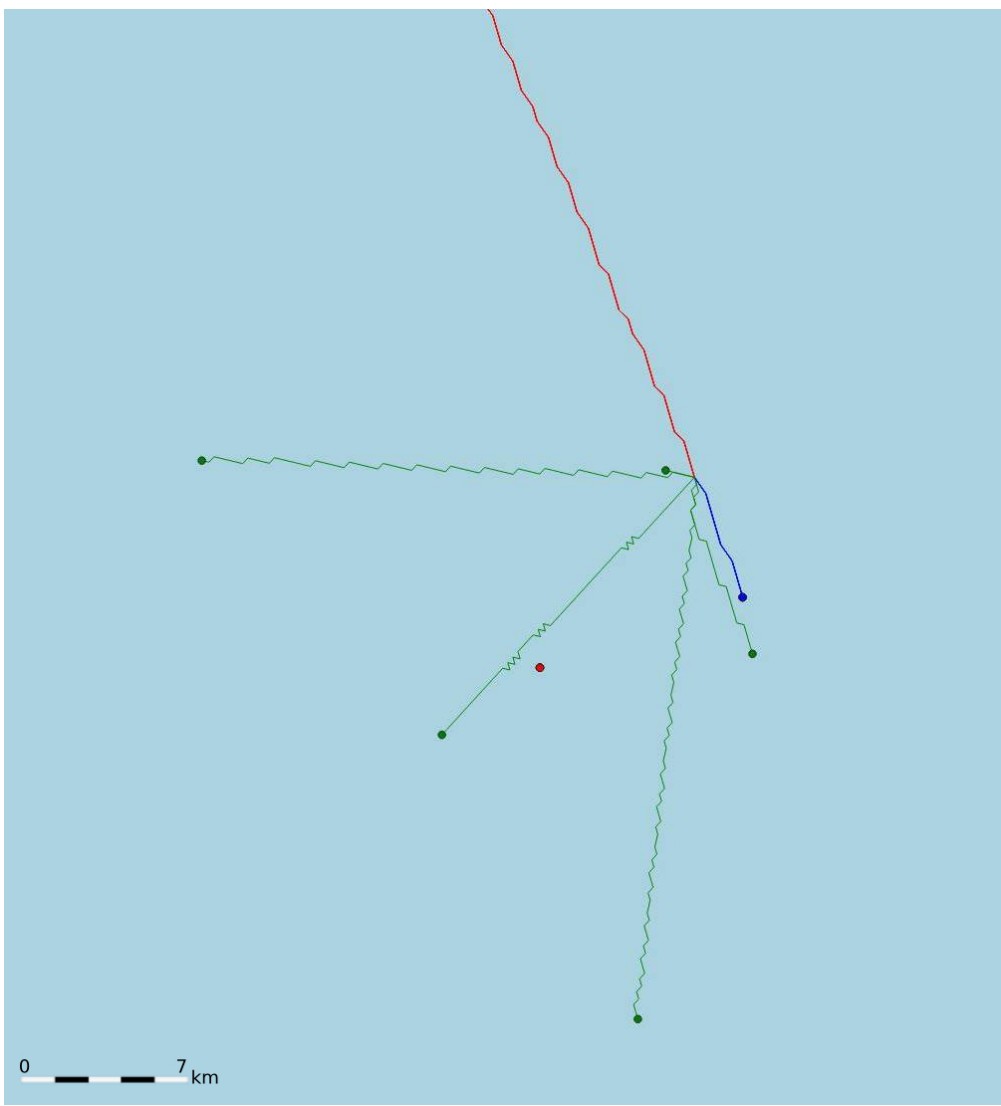

**Figure 18: More examples of hallucinations.** *The red point is the barycentre of the polygon formed by the green points at the ends of the green trajectories.*

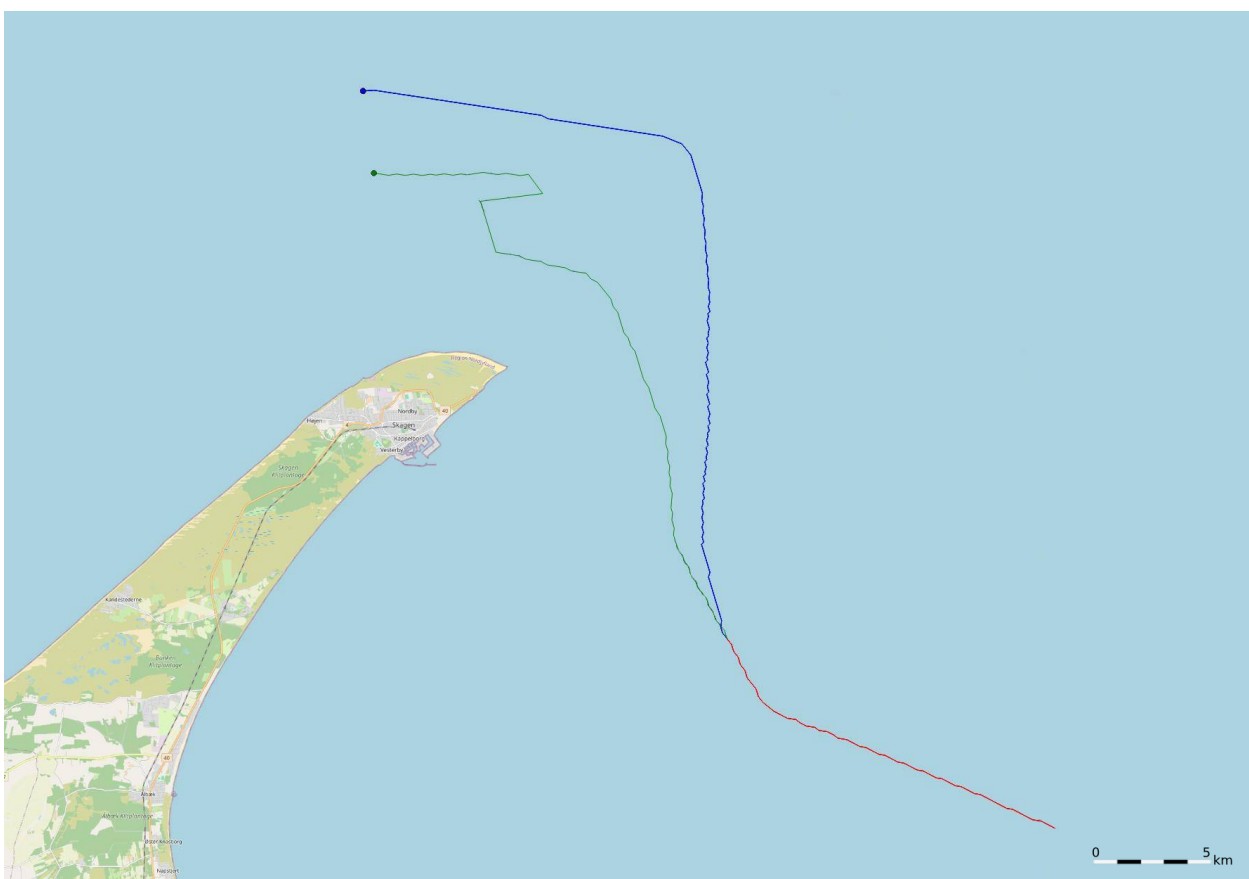

**Figure 19: More examples of hallucinations.**

