# OpenReview forum: "Enhancing Maritime Trajectory Forecasting via H3 Index and Causal Language Modelling (CLM)"
_TMLR — Accepted by TMLR_

### Review · Reviewer_Nwg3 · 2025-01-03

**Summary Of Contributions:**

1. This paper proposes an innovative approach combining Transformer architectures with spatial representation to address the omission of spatial features in traditional ship trajectory prediction methods, improving accuracy.
2. This paper utilises Uber's H3 spatial indexing system to discretise ship trajectories into hexagonal cells, resolving GNSS precision issues, and introduces a pseudo-octal H3 representation to enhance Transformer's spatial understanding and computational efficiency.
3. This paper explores the advantages of causal language models in ship trajectory prediction, leveraging their auto-regressive nature to capture temporal and causal structures, improving performance in complex maritime scenarios.

**Audience:**

Yes

**Broader Impact Concerns:**

It seems that this paper does not involve ethical implications.

**Claims And Evidence:**

Yes

**Requested Changes:**

1. It would be more logical to include a description of the causal language model in the introduction, even though a detailed explanation is provided in the later sections.
2. The model chooses Mixtral8x7B, and providing a simple architecture diagram for this model would help make the description clear.
3. If possible, a comparison with more existing methods should be included.

**Strengths And Weaknesses:**

Strengths
1. In terms of methodological innovation, to address the limitations of traditional methods in ship trajectory prediction, this study innovatively combines the Transformer architecture with spatial representation, using H3 spatial indexing and pseudo-octal representation to resolve GNSS precision issues.
2. This study directly addresses the practical challenges in maritime operations, offering significant practical value.

Weaknesses
1. The article title emphasizes the H3 index and causal language models, but the introduction lacks a description of the causal language model.
2. The article provides limited comparison between the proposed method and other existing methods.

---

> ### Author Response · Authors · 2025-01-21
> **Response to Reviewer Nwg3**
>
> We are grateful for the time you took to review our paper and for sharing your valuable feedback. Your observations have been important in highlighting areas for improvement. Below, we address your feedback and describe the changes we intend to implement:
>
> ---
>
> ### Introduction Update
>
> **Description of Causal Language Model (CLM)**
>
> We appreciate your suggestion to include a description of the causal language model in the introduction. Given that CLM is central to our methodology, we agree that it merits an introduction early in the paper.
>
> **Planned Changes:**
>
> - **CLM Description:** We will add a paragraph in the introduction section that highlights the role and advantages of causal language models in ship trajectory prediction. This will provide readers with a clear understanding of how CLM integrates with our innovative approach from the outset. We will insert it after the sentence: "_In doing so, we aim to address the limitations of existing models, making maritime operations more efficient, secure, and economically viable._", first sentence of page 2.
>
> **Suggested Change**
>
> _In this research, we adopt a Causal Language Model (CLM) due to its ability to effectively handle sequential dependencies, a crucial aspect of ship trajectory prediction. The auto-regressive nature of CLMs aligns naturally with the temporal progression of vessel movements, where each position is inherently influenced by prior locations. By leveraging the capacity of CLMs to predict future tokens based on preceding ones, we ensure that the spatial and temporal continuity of trajectories is accurately captured. This sequential modelling capability, coupled with our innovative spatial representation strategy, offers a robust framework for understanding and forecasting complex maritime patterns._
>
> ### Architecture Diagram for Mixtral8x7B
>
> **Clarity and Readability**
>
> We understand the importance of visual aids in explaining complex models. Including a detailed architecture diagram for Mixtral8x7B can indeed enhance clarity and readability.
>
> **Planned Changes:**
>
> - **Architecture Diagram:** We will include a simple and descriptive architecture diagram for Mixtral8x7B. This diagram will highlight the key components and their interactions, making it easier for readers to follow the model's structure and functioning.
>
> ---
>
> ### Comparison with Existing Methods
>
> **Benchmarking and State-of-the-Art Comparison**
>
> We recognize the importance of benchmarking our proposed method against existing approaches to demonstrate its effectiveness. We did not include the other methods as they are not autoregressive \[1, 2, 3\]. If it was unclear, we can specify it in the related work. Anyway, we built the Seq2Seq model and we can compare ourselves with it.
>
> **Planned Changes:**
>
> - **Comparative Analysis:** We will expand our comparative analysis section to include a Seq2Seq model and a modified one with GRU cells. This will provide a comprehensive evaluation of our approach, ensuring that our contributions are clearly articulated and validated.
>
>
> \[1\]: Cheng-Hong Yang, Chih-Hsien Wu, Jen-Chung Shao, Yi-Chuan Wang, and Chih-Min Hsieh. Ais-based intelligent vessel trajectory prediction using bi-lstm. IEEE Access, 10:24302–24315, 2022. doi: 10.1109/ACCESS.2022.3154812.
>
> \[2\]: Mingze Li, Bing Li, Zhigang Qi, Jiashuai Li, and Jiawei Wu. Enhancing maritime navigational safety: Ship trajectory prediction using acoatt–lstm and ais data. ISPRS International Journal of Geo-Information, 13 (3), 2024. ISSN 2220-9964. doi: 10.3390/ijgi13030085. URL https://www.mdpi.com/2220-9964/13/3/85.
>
> \[3\]: Kristian Aalling Sørensen, Peder Heiselberg, and Henning Heiselberg. Probabilistic maritime trajectory prediction in complex scenarios using deep learning. Sensors, 22(5), 2022. ISSN 1424-8220. doi: 10.3390/s22052058. URL https://www.mdpi.com/1424-8220/22/5/2058.

---

> > ### Comment · Reviewer_Nwg3 · 2025-01-24
> > **Official Comment by Reviewer Nwg3**
> >
> > My main concerns have been addressed, and the corresponding changes have been made in the revised paper, so I tend to accept this paper.

---

### Review · Reviewer_2Vsr · 2025-01-11

**Summary Of Contributions:**

This paper focuses on applying transformer to the prediction of maritime trajectories, presenting an application-oriented study that leverages established techniques in an innovative context. The contributions can be summarized as follows:
1. Interesting Application.
2. Methodology Based on Classic Model Mistral 8x7B.
3. Predictable but Useful Results.
4. Limited Comparison Against Existing Methods:

**Audience:**

Yes

**Claims And Evidence:**

Yes

**Requested Changes:**

1. Expand Generalization Analysis: Provide a detailed evaluation of the model’s performance across diverse geographic regions and maritime scenarios to validate its robustness and adaptability.
2. Address Motivation for Transformer Adoption: Include a comparison with LSTM or GRU-based models to demonstrate clear advantages (e.g., accuracy, long-term dependencies, computational efficiency).

**Strengths And Weaknesses:**

Strength:
1. Novel Research Topic: The focus on maritime trajectory forecasting, especially using GNSS data alone, is an uncommon and intriguing application in machine learning, providing valuable insights for specific domains like logistics and maritime safety.
2. Robust Methodology: The use of H3 geospatial indexing for discretizing spatial data is well-suited for this task, mitigating the challenges of GNSS inaccuracies and enabling hierarchical spatial representation. The model’s integration of classic transformer techniques with spatial data highlights practical applications of NLP methods in non-traditional fields.

Weakness:
1. Limited Benchmarking: The model comparison is limited to Kalman filters and TrAISformer. There is no comparison with traditional RNN-based methods, such as LSTM or GRU, commonly used for trajectory forecasting. This weakens the claim that the proposed approach is superior for this task
2. Insufficient Generalization Analysis: Although the model generalizes well globally, the lack of specific regional analysis or adaptability to diverse maritime environments limits insights into its robustness across unique operational settings.
3. Dependence on Data Availability: The reliance on proprietary AIS data limits reproducibility and wider adoption within the machine learning community. Without open datasets, the value of this work to the broader research community remains unclear.

---

> ### Author Response · Authors · 2025-01-21
> **Response to Reviewer 2Vsr**
>
> Thank you for your thorough review of our paper and for offering such feedback. Your suggestions have been essential in pinpointing areas where we can enhance our work. Below, we respond to your comments and outline the proposed changes:
>
> ---
>
> ### Comparison with Existing Methods
>
> **Incorporating More Comparison Methods**
>
> We understand the importance of providing a comprehensive comparison with existing methods to validate the superiority of our approach. We will incorporate more methods in our comparative analysis.
>
> Can you please clarify what you mean by _Long-Term Dependencies_?
>
> **Planned Changes:**
> - **Additional Methods:** We will include comparisons with traditional autoregressive RNN-based methods such as LSTM and GRU (Seq2Seq models), which are commonly used for trajectory forecasting.
> - **Computational Efficiency:** We will also include an analysis of the computational efficiency of our model compared to others, highlighting its resource usage and the type of hardware it can run on.
>
> ---
>
> ### Generalization Analysis
>
> **Evaluating Performance Across Diverse Regions**
>
> Your suggestion to provide a detailed evaluation of the model’s performance across diverse geographic regions is well-received. This will help validate the robustness and adaptability of our model. Initially, we used 100% of our dataset to train the model. Since AIS data can be captured live, we made this unusual choice. During the evaluation phase of our model, we used data recaptured 6 months later (in August). For the comparison with TrAISformer, we used the dataset from that author's method; however, our training set contained only a tiny fraction of it (around 2%). Additionally, we are going to include a geographical area that has never been seen before to conduct a zero-shot evaluation of the model.
>
> **Planned Changes:**
> - **Case Studies:** We will add a case study to demonstrate that our model performs well across different geographic areas without compromising performance. More specifically, We plan to add the region of Malacca Strait, Singapore Strait and South China Sea, as our model has never seen these data before.
> - **Rename the models** : We will rename our model's version "w/o finetuning" to "Few-Shot task"
>
> ---
>
> ### Data Availability
>
> **Addressing Reproducibility Concerns**
>
> We acknowledge the concern regarding the use of proprietary AIS data. Just to clarify it, while our data was collected using military equipment, similar AIS data is publicly available on platforms like Vessel Finder and Marine Traffic. The experiment is reproductible no matter where the data come from, the only thing to consider is to have the columns _lat_ and _lon_, and _SOG_ and _COG_ if your dataset is not cleaned.
>
> Public data are also available here:
>
> - https://www.aishub.net/
> - https://www.kystverket.no/en/navigation-and-monitoring/ais/access-to-ais-data/
>
>
> **Planned Changes**
> - We will add the links in the Data Availability section.

---

### Review · Reviewer_1ide · 2025-01-15

**Summary Of Contributions:**

Ship trajectory prediction is an evolving area in AI, with traditional methods relying on techniques like LSTMs, GRUs, and Transformers. This study offers an alternative approach, predicting trajectories using only GNSS positions by treating the problem as a natural language processing task. AIS latitude/longitude data is converted into cell identifiers via the H3 index, making it easier for language models to capture spatial patterns. The Fréchet distance is used for evaluation and results demonstrate the promise of this solution against an established baseline.

**Audience:**

Yes

**Claims And Evidence:**

Yes

**Requested Changes:**

- Additional baselines are needed to understand improvements over SOTA methods

The paper is lacking in terms of comparisons with baselines. From traditional forecasting methods to modern DNN and Foundation Models, such methods have to be included to understand the improvement over SOTA methods.

- Discussion about the scalability of the solution is also necessary considering the need to run such models in streaming manner

In practice, such methods have to be applied in real-time and therefore, some discussion and experiments (vs. baselines) are needed to understand the computational cost of this solution.

- The extension of H3 lacks technical depth but seems to work reasonably well

The solution is mainly a simple extension of H3. It seems to work and improve performance, yet, it lacks technical depth (it appears as a simple transformation step, which is unclear if this has to appear as a full paper). Novelty is somewhat low for this work.

**Strengths And Weaknesses:**

Strengths:

- Ship trajectory mining is an important AI area with a lot of promise and applications
- The discretization step is an essential tool to model patterns and improve prediction
- Experimental results show the promise of the solution with qualitative and quantitative results

Weaknesses:

- Additional baselines are needed to understand improvements over SOTA methods
- Discussion about the scalability of the solution is also necessary considering the need to run such models in streaming manner
- The extension of H3 lacks technical depth but seems to work reasonably well

---

> ### Author Response · Authors · 2025-01-21
> **Response to Reviewer 1ide**
>
> We deeply appreciate the time and effort you've invested in reviewing our paper and the constructive feedback you've provided. Your comments have been helpful in identifying areas where we can make improvements. Below, we address each of your points and detail the changes we plan to make:
>
> ---
>
> ### Additional Baselines and Comparisons
>
> We understand the importance of comparing our work with other methods. We initially considered other works but excluded them from our study because they were not autoregressive. If this is not clear, we will clarify this point in the Related Work section. Additionally, could you please clarify what you mean by "Foundation Models"? We are not sure which specific models you are referring to.
>
> **Planned Changes:**
> -  We will extend the Related Work section in order to explain why we excluded non-autoregressive methods from our study. This will provide readers with a clearer understanding of our methodological choices.
> -  We will add a comparison with a Seq2Seq model based on Sutskever's work and a GRU-based variant in our comparisons.
>
>
> ### Scalability and Real-Time Application:
>
> Our model is not designed for real-time streaming applications like a chatbot. Instead, it is intended for use with manual user actions in GIS software like QGIS. However, we will add a comparison of the model's computational cost using the FLOPS metric and the number of model parameters to address your concern about scalability.
>
> **Planned Changes:**
> - **Computational Cost Analysis:** We will include a comparison of the model's computational cost using the FLOPS metric and the number of model parameters. This will help readers understand the scalability and efficiency of our approach.
> - **Clarification on Real-Time Application:** We will add a note in the Conclusion section clarifying that our model is designed for manual user actions in GIS software, not for real-time streaming applications.
>
> ### Extension of H3:
>
> The H3 usage is not the core of our method, although it plays an important role by adding a hierarchical spatial structure. The main focus of our paper is using a Large Language Model (LLM) to process a time-series task. It is not just a use of a transformer architecture, which would add attention to a time-series problem. Instead, it is a concept in which we consider that it is possible to represent these data sequences in a simple 'natural' language with a grammar specific to our task, and we use this LLM to continue the sequences. Here, the grammar is provided by using the H3 spatial structure, which we modified for our task. Moreover, in the Introduction section, we mention another study that used a similar method, where they use hexagonal cells instead of the H3 Index.
>
> **Planned Changes:**
> - **Clarification on H3 Usage:** We will emphasize that H3 is a tool used to add hierarchical spatial structure and is not the core of our method. This will help readers understand the role of H3 in our approach more clearly.
>
> Thank you once again for your valuable feedback. We will incorporate your suggestions to improve the paper.

---

### Author Response · Authors · 2025-01-22
**Summaries of revisions during the review period**

Dear Reviewers,

Thank you for your time and effort in reviewing our paper. Your feedback has been helping us improve the clarity, accuracy, and overall quality of our work. We have carefully addressed all of your comments and made several revisions to enhance the manuscript. Below, we have summarized the key changes made in response to your suggestions :

# Changes

- **Abstract Updated**: The abstract has been revised to explicitly state the comparison with Seq2Seq and TrAISformer models.
- **Introduction Enhanced**: An additional paragraph has been added to the introduction (page 2) to introduce Causal Language Models.
- **H3 Index Clarified**: Another paragraph has been added to the introduction (page 2) to clarify the usage of the H3 index, not as an extension of H3 but as a tool into a more valuable concept.
- **Related Work Section Updated**: A clarification has been added to the Related Work section (page 5) regarding non-autoregressive methods.
- **Architecture Diagram Added**: A schematic diagram of the architecture has been included in section 4.2.1 (page 9) and referenced in the text on page 10.
- **State-of-the-Art Comparison Improved**: The comparison with the state of the art in Table 3 (page 15) has been enhanced for clarity, comparison against Seq2Seq models (LSTM and GRU versions).
- **Case Study Added**: A case study on an unseen region has been added, reported in Table 3 and explained in the text on page 17 in section 5.1. The unseen region is the Singapore Strait, as there is a lot of vessels with complex trajectories.
- **Clarification in Conclusion**: The usage of our method in non-streaming scenarios has been clarified in the conclusion. We designed this model to be used in GIS software and not in a chatbot-like application.
- **Data Availability Section Updated**: Links to the public AIS datasets have been added to the Data Availability section (page 19).

Thank you again for your contribution to the improvement of our research.

Best regards,

---

### Decision · Action_Editor_mZwp · 2025-03-10

**Recommendation:** Accept with minor revision

**Comment:**

When responding to the reproducibility concerns raised by reviewer 2Vsr, the authors pointed out that there exist free, publicly available AIS datasets. While adding links in the Data Availability section would help, greater impact could be made if even some preliminary experiments using these datasets would be added to the paper to demonstrate the generalization capabilities of the proposed method.

**Audience:**

Maritime trajectory forecasting probably has a narrow audience in the TMLR community. However, the study may also be relevant to many other spatiotemporal forecasting tasks which have growing interests in machine learning.

**Claims And Evidence:**

This paper explores the potential of an alternative approach in predicting ship trajectories. Instead of the traditional approach based on such models as LSTM, GRU, and transformers for spatiotemporal forecasting, this study uses only GNSS positions and regards the spatiotemporal forecasting problem as similar to a natural language processing task that can be solved using a causal language model.

The prediction performance of the proposed method was verified empirically. However, all reviewers had concerns about the original version of the submitted paper in that, among other things, some baselines were not included in the comparative study. We thank the authors for including more experiments in their revised paper to address the comments. With the revisions, the claims are now more convincing.